# EXPLAIN YOURSELF, BRIEFLY!
## SELF-EXPLAINING NEURAL NETWORKS WITH CONCISE SUFFICIENT REASONS

**Shahaf Bassan**[1,2], **Ron Eliav**[1,3], **Shlomit Gur**[1]
IBM Research[1], The Hebrew University of Jerusalem[2], Bar-Ilan University[3]
`shahaf.bassan@mail.huji.ac.il`

## ABSTRACT

*Minimal sufficient reasons* represent a prevalent form of explanation — the smallest subset of input features which, when held constant at their corresponding values, ensure that the prediction remains unchanged. Previous *post-hoc* methods attempt to obtain such explanations but face two main limitations: (i) Obtaining these subsets poses a computational challenge, leading most scalable methods to converge towards suboptimal, less meaningful subsets; (ii) These methods heavily rely on sampling out-of-distribution input assignments, potentially resulting in counterintuitive behaviors. To tackle these limitations, we propose in this work a self-supervised training approach, which we term *sufficient subset training* (SST). Using SST, we train models to generate concise sufficient reasons for their predictions as an integral part of their output. Our results indicate that our framework produces succinct and faithful subsets substantially more efficiently than competing post-hoc methods, while maintaining comparable predictive performance. Code is available at: `https://github.com/IBM/SAX/tree/main/ICLR25`.

## 1 INTRODUCTION

Despite the widespread adoption of deep neural networks, understanding how they make decisions remains challenging. Initial attempts focused on additive feature attribution methods (Ribeiro et al. (2016); Lundberg & Lee (2017); Sundararajan et al. (2017)), which often treat the model's behavior as approximately linear around the input of interest. A different line of techniques, such as Anchors (Ribeiro et al. (2018)) and SIS (Carter et al. (2019)), seek to pinpoint *sufficient* subsets of input features that ensure that a prediction remains unchanged. Here, the goal often involves finding the *smallest* possible sufficient subset, which is typically assumed to provide a better interpretation (Ribeiro et al. (2016); Carter et al. (2019); Ignatiev et al. (2019); Barceló et al. (2020); Wäldchen et al. (2021)).

Building on this foundation, we adopt the common notation of a *sufficient reason* (Barceló et al. (2020); Darwiche & Hirth (2020); Arenas et al. (2022)) to refer to any subset of input features $S$ that ensures the prediction of a classifier remains constant. We categorize previous approaches into three broad categories of sufficiency, namely: (i) *baseline* sufficient reasons, which involve setting the values of the complement $\overline{S}$ to a fixed baseline; (ii) *probabilistic* sufficient reasons, which entail sampling $\overline{S}$ from a specified distribution; and (iii) *robust* sufficient reasons, which aim to confirm that the prediction does not change for any possible assignment of values to $\overline{S}$ within a certain domain.

Several methods have been proposed for obtaining sufficient reasons for ML classifiers (baseline, probabilistic, and robust) (Ribeiro et al. (2018); Carter et al. (2019); Ignatiev et al. (2019); Wang et al. (2021a); Chockler et al. (2021)). These methods are typically *post-hoc*, meaning their purpose is to interpret the model's behavior *after* training. There are two major issues concerning such post-hoc approaches: (i) First, identifying cardinally minimal sufficient reasons in neural networks is computationally challenging (Barceló et al. (2020); Wäldchen et al. (2021)), making most methods either unscalable or prone to suboptimal results. (ii) Second, when assessing if a subset $S$ is a sufficient reason, post-hoc methods typically fix $S$'s values and assign different values to the features of $\overline{S}$. This mixing of values may result in *out-of-distribution* (OOD) inputs, potentially misleading

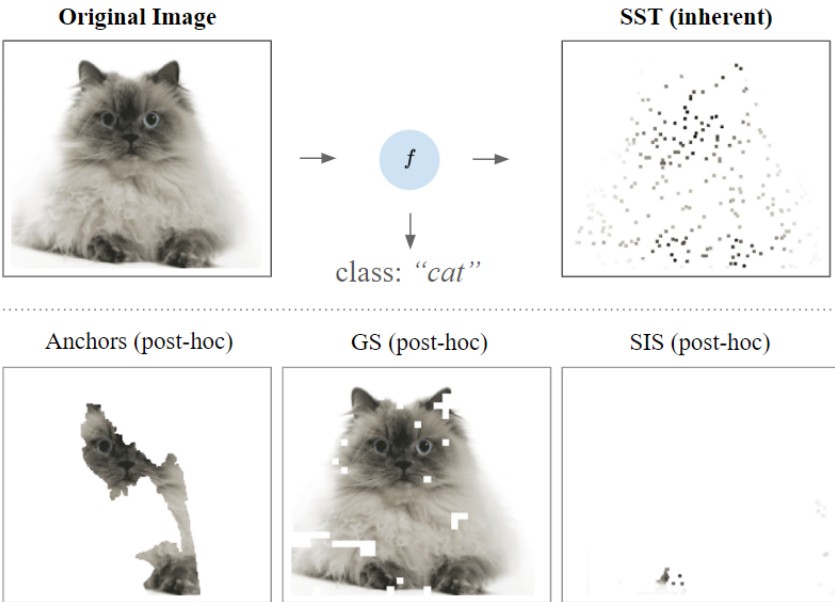

Figure 1: An example of a sufficient reason generated by a model trained with *sufficient subset training* (SST) on the IMAGENET dataset, compared to those generated by post-hoc methods on standard-trained models. While explanations from Anchors and GS are larger, those from SIS are less faithful and lack subset sufficiency (details in the experiments in Section 5). SST generates explanations that are both concise and faithful, while performing this task with significantly improved efficiency. Additional examples appear in appendix J.

the evaluation of $S$'s sufficiency. This issue has been identified as a significant challenge in such methods (Hase et al. (2021); Vafa et al. (2021); Yu et al. (2023); Amir et al. (2024)), frequently leading to the identification of subsets that do not align with human intuition (Hase et al. (2021)).

**Our contributions.** We start by addressing the previously mentioned intractability and OOD concerns, reinforcing them with new computational complexity findings on the difficulty of generating such explanations: (i) First, we extend computational complexity results that were limited only to the *binary* setting (Barceló et al. (2020); Wäldchen et al. (2021)) to any discrete or *continuous* domain, showcasing the intractability of obtaining cardinally minimal explanations for a much more general setting; (ii) Secondly, we prove that the intractability of generating these explanations holds for different *relaxed* conditions, including obtaining cardinally minimal *baseline* sufficient reasons as well as *approximating* the size of cardinally minimal sufficient reasons. These results reinforce the computational intractability of obtaining this form of explanation in the classic post-hoc fashion.

Aiming to mitigate the intractability and OOD challenges, we propose a novel self-supervised approach named *sufficient subset training (SST)*, designed to inherently produce concise sufficient reasons for predictions in neural networks. SST trains models with two outputs: (i) the usual predictive output and (ii) an extra output that facilitates the extraction of a sufficient reason for that prediction. The training integrates two additional optimization objectives: ensuring the extracted subset is sufficient concerning the predictive output and aiming for the subset to have minimal cardinality.

We integrate these objectives using a dual propagation through the model, calculating two additional loss terms. The first, the *faithfulness loss*, validates that the chosen subset sufficiently determines the prediction. The second, the *cardinality loss*, aims to minimize the subset size. We employ various masking strategies during the second propagation to address three sufficiency types. These include setting $\overline{S}$ to a constant baseline for baseline sufficient reasons, drawing $\overline{S}$ from a distribution for probabilistic sufficient reasons, or using an adversarial attack on $\overline{S}$ for robust sufficient reasons.

Training models with SST reduces the need for complex post-hoc computations and exposes models to relevant mixed inputs, thereby reducing the sensitivity of subsets to OOD instances. Moreover,

our method's versatility enables its use across any neural network architecture. We validate SST by testing it on multiple architectures for image and language classification tasks, using diverse masking criteria. We compare our training-generated explanations with those from popular post-hoc methods on standard (i.e., classification only) models. Our results show that SST produces sufficient reasons that are concise and faithful, more efficiently than post-hoc approaches, while preserving comparable predictive performance. Figure 1 illustrates an example of a sufficient reason generated by SST, in comparison to post-hoc explanations.

## 2 FORMS OF SUFFICIENT REASONS

In this work, our focus is on *local* interpretations of classification models. Specifically, for a model $f : \mathbb{R}^n \to \mathbb{R}^c$ and an input $\mathbf{x} \in \mathbb{R}^n$, we aim to explain the prediction $f(\mathbf{x})$. Formally, we define a *sufficient reason* (Barceló et al. (2020); Darwiche & Hirth (2020); Arenas et al. (2022)) as any subset of features $S \subseteq [n]$, such that fixing $S$ to their values in $\mathbf{x}$, determines the prediction $f(\mathbf{x})$. We specifically want the prediction $f(\mathbf{x}_S; \mathbf{z}_{\bar{S}})$ to align with $f(\mathbf{x})$, where $(\mathbf{x}_S; \mathbf{z}_{\bar{S}})$ denotes an input for which the values of $S$ are drawn from $\mathbf{x} \in \mathbb{R}^n$ and the values of $\overline{S}$ are drawn from $\mathbf{z} \in \mathbb{R}^n$. If $S$ includes all input features, i.e., $S := \{1, \ldots, n\}$, it is trivially sufficient. However, it is common to seek more concise sufficient reasons (Ribeiro et al. (2018); Carter et al. (2019); Ignatiev et al. (2019); Barceló et al. (2020); Blanc et al. (2021); Wäldchen et al. (2021)). Another consideration is the choice of values for the complement set $\overline{S}$. We classify previous methods into three general categories:

**Definition 1.** *Given a model $f$, an instance $\mathbf{x} \in \mathbb{R}^n$ and some baseline $\mathbf{z} \in \mathbb{R}^n$, we define $S \subseteq \{1, \ldots, n\}$ as a* baseline sufficient reason *of $\langle f, \mathbf{x} \rangle$ with respect to $\mathbf{z}$ iff it holds that:*

$$\arg\max_j \ f(\mathbf{x}_S; \mathbf{z}_{\bar{S}})_j = \arg\max_j \ f(\mathbf{x})_j \tag{1}$$

The baseline $\mathbf{z}$ represents the "missingness" of the complementary set $\overline{S}$, but its effectiveness is highly dependent on choosing a *specific* $\mathbf{z}$. While it is feasible to use a small number of different baselines to mitigate this dependency, such an approach is not practical across broader domains. However, in certain cases, employing baselines may be intuitive, especially when they carry specific significance in that scenario. For instance, in the context of language classification tasks where $f$ represents a transformer architecture, $\mathbf{z}$ could be assigned the widely recognized MASK token, which is utilized during pre-training phases to address the concept of "missingness" in tokens (Devlin et al. (2018)).

Conversely, sufficiency can imply that the values of $S$ dictate the prediction, regardless of *any* assignment to $\overline{S}$ in some domain. This definition is particularly useful in cases where a strict sufficiency of $S$ is imperative, such as in safety-critical systems (Marques-Silva & Ignatiev (2022); Ignatiev et al. (2019); Wu et al. (2024b)). There, the presence of any counterfactual assignment over $\overline{S}$ may be crucial. While in some instances the domain of $\overline{S}$ is unlimited (Ignatiev et al. (2019); Marques-Silva & Ignatiev (2022)), a more feasible modification is to consider a limited continuous domain surrounding $\mathbf{x}$ (Wu et al. (2024b); La Malfa et al. (2021); Izza et al. (2024)). We define this kind of sufficiency rationale as *robust* sufficient reasons:

**Definition 2.** *Given a model $f$ and an instance $\mathbf{x} \in \mathbb{R}^n$, we define $S \subseteq \{1, \ldots, n\}$ as a* robust sufficient reason *of $\langle f, \mathbf{x} \rangle$ on an $\ell_p$-norm ball $B_p^{\epsilon_p}$ of radius $\epsilon_p$ iff it holds that:*

$$\forall \mathbf{z} \in B_p^{\epsilon_p}(\mathbf{x}). \quad [\arg\max_j \ f(\mathbf{x}_S; \mathbf{z}_{\bar{S}})_j = \arg\max_j \ f(\mathbf{x})_j],$$
$$s.t \quad B_p^{\epsilon_p}(\mathbf{x}) := \{\mathbf{z} \in \mathbb{R}^n | \ ||\mathbf{x} - \mathbf{z}||_p \leq \epsilon_p\} \tag{2}$$

However, it is not always necessary to assume that fixing the subset $S$ guarantees a consistent prediction across an entire $B_p^{\epsilon_p}(\mathbf{x})$ domain. Rather, when $S$ is fixed, the classification may be maintained with a certain probability (Ribeiro et al. (2018); Blanc et al. (2021); Wäldchen et al. (2021); Wang et al. (2021a)). We refer to these as *probabilistic* sufficient reasons. Under this framework, fixing $S$ and sampling values from $\overline{S}$ according to a given distribution $\mathcal{D}$ ensures that the classification remains unchanged with a probability greater than $1 - \delta$:

**Definition 3.** *Given a model $f$ and some distribution $\mathcal{D}$ over the input features, we define $S \subseteq \{1, \ldots, n\}$ as a* probabilistic sufficient reason *of $\langle f, \mathbf{x} \rangle$ with respect to $\mathcal{D}$ iff:*

$$\Pr_{\mathbf{z} \sim \mathcal{D}} [\arg\max_j \ f(\mathbf{z})_j = \arg\max_j \ f(\mathbf{x})_j \mid \mathbf{z}_S = \mathbf{x}_S] \geq 1 - \delta \tag{3}$$

where $\mathbf{z}_S = \mathbf{x}_S$ denotes fixing the features of $S$ in the vector $\mathbf{z}$ to their corresponding values in $\mathbf{x}$.

## 3 PROBLEMS WITH POST-HOC MINIMAL SUFFICIENT REASON COMPUTATIONS

**Intractability.** Obtaining cardinally minimal sufficient reasons is known to be computationally challenging (Barceló et al. (2020); Wäldchen et al. (2021)), particularly in neural netowrks (Barceló et al. (2020); Adolfi et al. (2024); Bassan et al. (2024)). Two key factors influence this computational challenge — the first is finding a *cardinally minimal* sufficient reason, which may be intractable when the input space is large. This tractability barrier is common to all of the previous forms of sufficiency (baseline, probabilistic, and robust). A second computational challenge can emerge when verifying the sufficiency of even *a single* subset proves to be computationally difficult. For instance, in *robust* sufficient reasons, confirming subset sufficiency equates to robustness verification, a process that by itself is computationally challenging (Katz et al. (2017); Sälzer & Lange (2021)). The intractability of obtaining cardinally minimal sufficient reasons is further exemplified by the following theorem:

**Theorem 1.** *Given a neural network classifier $f$ with ReLU activations and $\mathbf{x} \in \mathbb{R}^n$, obtaining a cardinally minimal sufficient reason for $\langle f, \mathbf{x} \rangle$ is (i) NP-Complete for baseline sufficient reasons (ii) $\Sigma_2^P$-Complete for robust sufficient reasons and (iii) $NP^{PP}$-Hard for probabilistic sufficient reasons.*

The highly intractable complexity class $\Sigma_2^P$ encompasses problems solvable in NP time with constant-time access to a coNP oracle, whereas $NP^{PP}$ includes problems solvable in NP time with constant-time access to a probabilistic PP oracle. More details on these classes are available in Appendix B. For our purposes, it suffices to understand that $NP \subsetneq \Sigma_2^P$ and $NP \subsetneq NP^{PP}$ is widely accepted (Arora & Barak (2009)), highlighting their substantial intractability (problems that are significantly harder than NP-Complete problems).

*Proof sketch.* The full proof is relegated to Appendix C. We begin by extending the complexity results for *robust* and *probabilistic* sufficient reasons provided by (Barceló et al. (2020); Wäldchen et al. (2021)), which focused on either CNF classifiers or multi-layer perceptrons with *binary* inputs and outputs. We provide complexity proofs for a more general setting considering neural networks over any discrete or *continuous* input or output domains. Membership outcomes to continuous domains stem from realizing that instead of guessing certificate assignments for binary inputs, one can guess the activation status of any ReLU constraint. For hardness, we perform a technical reduction that constructs a neural network with a strictly stronger subset sufficiency. We also provide a novel complexity proof, which indicates the intractability of the *relaxed* version of computing cardinally minimal *baseline* sufficient reasons, via a reduction from the classic CNF-SAT problem.

To further emphasize the hardness of the previous results, we prove that their *approximation* remains intractable. We establish this through approximation-preserving reductions from the Shortest-Implicant-Core (Umans (1999)) and Max-Clique (Håstad (1999)) problems (proofs provided in Appendix D):

**Theorem 2.** *Given a neural network $f$ with ReLU activations and $\mathbf{x} \in \mathbb{R}^n$, (i) $\forall \epsilon > 0$ approximating cardinally minimal robust sufficient reasons with $n^{\frac{1}{2}-\epsilon}$ factor is $\Sigma_2^P$-Hard, (ii) $\forall \epsilon > 0$ approximating cardinally minimal probabilistic or baseline sufficient reasons with factor $n^{1-\epsilon}$ is NP-Hard.*

These theoretical limitations highlight the challenges associated with achieving exact computations or approximations and do not directly apply to the practical execution time of methods that employ heuristic approximations. Nevertheless, many such methods, including Anchors (Ribeiro et al. (2018)) and SIS (Carter et al. (2019)), face issues with scalability and often rely on significant assumptions. For instance, these methods may substantially reduce the input size to improve the tractability of features (Ribeiro et al. (2018); Carter et al. (2019)) or concentrate on cases with a large margin between the predicted class's output probability and that of other classes (Carter et al. (2019)).

**Sensitivity to OOD counterfactuals.** Methods computing concise sufficient reasons often use algorithms that iteratively test inputs of the form $(\mathbf{x}_S; \mathbf{z}_{\bar{S}})$, comparing the prediction of $f(\mathbf{x}_S; \mathbf{z}_{\bar{S}})$

with $f(\mathbf{x})$ to determine $S$'s sufficiency (Ribeiro et al. (2018); Carter et al. (2019); Chockler et al. (2021)). A key issue is that $(\mathbf{x}_S; \mathbf{z}_{\bar{S}})$ might be OOD for $f$, potentially misjudging S's sufficiency. Iterative approaches aimed at finding *cardinally minimal* subsets may exacerbate this phenomenon, leading to significantly suboptimal or insufficient subsets. A few recent studies focused on NLP domains (Hase et al. (2021); Vafa et al. (2021)), aimed to enhance model robustness to OOD counterfactuals by arbitrarily selecting subsets $S$ and exposing models to inputs of the form $(\mathbf{x}_S; \mathbf{z}_{\bar{S}})$ during training. However, this arbitrary masking process lacks clarity on two important questions — (i) *which* features should be masked, and (ii) how *many* features should be masked . This, in turn, poses a risk of inadvertently masking critical features and potentially impairing model performance.

## 4 SELF-EXPLAINING NEURAL NETWORKS WITH SUFFICIENT REASONS

To address the earlier concerns, we train self-explaining neural networks that inherently provide concise sufficient reasons along with their predictions. This eliminates the need for costly post-computations and has the benefit of exposing models to OOD counterfactuals during training.

### 4.1 LEARNING MINIMAL SUFFICIENT REASONS

Our model is trained to produce two outputs: the initial prediction and an explanation vector that pinpoints a concise sufficient reason for that prediction. Each element in the vector corresponds to a feature. Through a Sigmoid output layer, we select a feature subset $S$ where outputs surpass a threshold $\tau$. More formally, we learn some: $h : \mathbb{R}^n \to \mathbb{R}^c \times [0, 1]^n$ where for simplicity we denote: $h(\mathbf{x}) := (h_1(\mathbf{x}), h_2(\mathbf{x}))$, for which $h_1$ denotes the *prediction* component and $h_2$, derived from a final Sigmoid layer, denotes the *explanation* component; both share the same hidden layers. The sufficient reason is taken by identifying $S := \{i \mid h_2(\mathbf{x})_i \geq \tau\}$. To ensure $S$ is sufficient, we perform a dual propagation through $h$, wherein the second forward pass, a new *masked input* is constructed by fixing the features in $S$ to their values in $\mathbf{x}$ and letting the features in $\bar{S}$ take some other values $\mathbf{z}$. The masked input $(\mathbf{x}_S; \mathbf{z}_{\bar{S}})$ is then propagated through $h$. This process is illustrated in Figure 2.

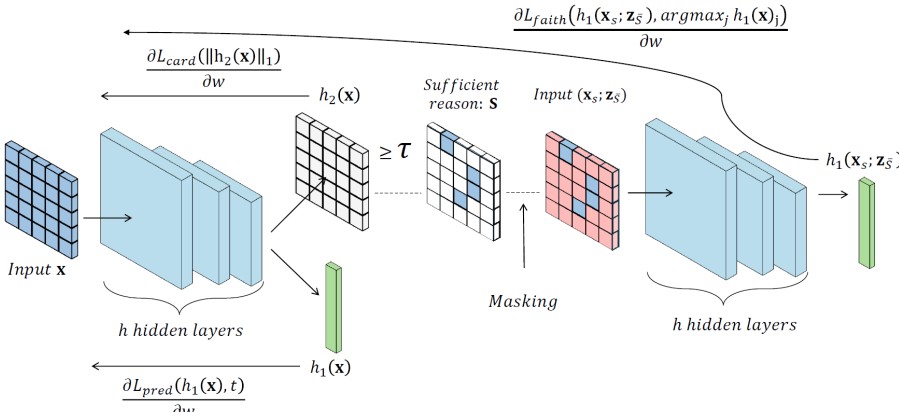

Figure 2: An illustration of the dual propagation incorporated during sufficient subset training

Besides the prediction task, we optimize two additional objectives to ensure that $S$ is both sufficient and concise. Our training incorporates three distinct loss terms: (i) the prediction loss, $L_{pred}$, (ii) a faithfulness loss, $L_{faith}$, enhancing the sufficiency of $S$ concerning $\langle h_1, \mathbf{x} \rangle$, and a cardinality loss, $L_{card}$, minimizing the *size* of $S$. We optimize these by minimizing the combined loss:

$$L_\theta(h) := L_{pred}(h) + \lambda L_{faith}(h) + \xi L_{card}(h) \tag{4}$$

Here, we use the standard cross entropy (CE) loss as our *prediction loss*, i.e., $L_{pred}(h) := L_{CE}(h_1(\mathbf{x}), t)$, where $t$ is the ground truth for $\mathbf{x}$. We use CE also in the *faithfulness loss*, which ensures the sufficiency of $S$ by way of minimizing the discrepancy between the predicted probabilities of the masked input $h_1(\mathbf{x}_S; \mathbf{z}_{\bar{S}})$ and the first prediction $h_1(\mathbf{x})$:

$$L_{faith}(h) := L_{CE}(h_1(\mathbf{x}_S; \mathbf{z}_{\bar{S}}), \arg\max_j h_1(\mathbf{x})_j), \ \ s.t \ \ S := \{i \mid h_2(\mathbf{x})_i \geq \tau\} \tag{5}$$

Training a model solely to optimize prediction accuracy and faithfulness may often result in larger sufficient reasons, as larger subsets are more likely to be sufficient. Specifically, the largest subset $S := \{1, \ldots, n\}$ always qualifies as sufficient, irrespective of the model or instance. Conversely, excessively small subsets may compromise prediction performance, leading to a bias towards larger subsets when these are the only objectives considered.

Hence, to train our model to produce *concise* sufficient reasons, we must include a third optimization objective aimed at minimizing the size of $S$. We incorporate this by implementing a *cardinality loss* which encourages the sparsity of $S$, by utilising the standard $L_1$ loss, i.e., $L_{card}(h) := ||h_2(\mathbf{x})||_1$.

### 4.2 MASKING

As noted earlier, during the model's second propagation phase, a masked input $(\mathbf{x}_S; \mathbf{z}_{\bar{S}})$ is processed by our model's prediction layer $h_1$. The faithfulness loss aims to align the prediction of $h_1(\mathbf{x}_S; \mathbf{z}_{\bar{S}})$ closely with $h_1(\mathbf{x})$. A key question remains on how to select the $\mathbf{z}$ values for the complementary features $\overline{S}$. We propose several masking methods that correspond to the different forms of sufficiency:

1. **Baseline Masking.** Given a fixed baseline $\mathbf{z} \in \mathbb{R}^n$, we fix the features of $\overline{S}$ to their corresponding values in $\mathbf{z}$ and propagate $h_1(\mathbf{x}_S; \mathbf{z}_{\bar{S}})$, optimizing $S$ to constitute as a *baseline* sufficient reason with respect to $\mathbf{z}$.

2. **Probabilistic Masking.** Given some distribution $\mathcal{D}$, we sample the corresponding assignments from $\mathcal{D}(\mathbf{z}|\mathbf{z}_S = \mathbf{x}_S)$ and propagate $h_1(\mathbf{x}_S; \mathbf{z}_{\bar{S}})$, optimizing $S$ to constitute as a *probabilistic* sufficient reason concerning $\mathcal{D}$.

3. **Robust Masking.** To ensure the sufficiency of $S$ across the *entire* input range of $B_p^{\epsilon_p}(\mathbf{x})$, we conduct an adversarial attack on $\mathbf{x}$. Here, we fix the values of $S$ and perturb only the features in $\overline{S}$. This approach involves identifying the "hardest" instances, denoted by $(\mathbf{x}_S'; \mathbf{z}_{\bar{S}}') \in B_p^{\epsilon_p}(\mathbf{x}_S; \mathbf{z}_{\bar{S}})$, and optimizing the prediction of $h_1(\mathbf{x}_S'; \mathbf{z}_{\bar{S}}')$ to assimilate that of $h_1(\mathbf{x})$. We achieve this by initially setting $(\mathbf{x}_S; \mathbf{z}_{\bar{S}})_0$ randomly within $B_p^{\epsilon_p}(\mathbf{x}_S; \mathbf{z}_{\bar{S}})$ and updating it over $N$ projected gradient descent (PGD) steps (Madry et al. (2017)):

$$(\mathbf{x}_S; \mathbf{z}_{\bar{S}})_{n+1} = \Pi_{B^\epsilon(\mathbf{x})}(\mathbf{x}_S; \mathbf{z}_{\bar{S}})_n + \alpha \cdot \text{sign}(\nabla_{(\mathbf{x}_S; \mathbf{z}_{\bar{S}})_n} L_{pred}(h_1(\mathbf{x}_S; \mathbf{z}_{\bar{S}})_n, t)) \tag{6}$$

where $\alpha$ is the step size, and $\Pi$ is the projection operator. This process is similar to adversarial training (Shafahi et al. (2019)), with the difference of not perturbing the features in $S$.

## 5 EXPERIMENTS

We assess SST across image and language classification tasks, comparing the sufficient reasons generated by our approach to those from state-of-the-art *post-hoc* methods. Following common conventions (Wu et al. (2024b); Ignatiev et al. (2019); Bassan & Katz (2023)), we evaluate these subsets on three metrics: (i) the mean *time* taken to generate a sufficient reason, (ii) the mean *size* of the sufficient reasons, and (iii) the mean *faithfulness*, i.e., how sufficient the subsets are. Particularly, when evaluating faithfulness, we categorize the results into three distinct types of sufficiency: baseline, probabilistic, and robust. Each category measures the proportion of test instances where the derived subset $S$ remains sufficient under baseline, probabilistic, or robust masking.

### 5.1 SUFFICIENT SUBSET TRAINING FOR IMAGES

We train SST-based models on three prominent image classification tasks: MNIST (Deng (2012)) digit recognition using a feed-forward neural network, CIFAR-10 (Krizhevsky et al. (2009)) using Resnet18 (He et al. (2016)), and IMAGENET using a pre-trained Resnet50 model (He et al. (2016)). To simplify the training process, we set the threshold at $\tau = 0.5$ and the faithfulness coefficient at $\lambda = 1$, conducting grid search exclusively over the cardinality coefficient $\xi$. Full training details can be found in Appendix E. We assess the inherent sufficient reasons generated by SST and

compare them with those from standard-trained models using post-hoc methods: *Anchors* (Ribeiro et al. (2018)), *SIS* (Carter et al. (2019)), and a vision-based technique (Fong & Vedaldi (2017)) which we term *gradient search* (*GS*).

**Robust Sufficient Reasons in Images.** In image classification tasks, it is common to evaluate the *robust* sufficiency of the identified sufficient reasons, by ensuring they remain sufficient across a continuous domain (Wu et al. (2024b); Ignatiev et al. (2019); Bassan & Katz (2023)). We aim to explore the ability of SST-based models to generate such subsets, initially training all models using robust masking with a PGD $\ell_\infty$ ($\epsilon = 0.12$) attack on the features in $\bar{S}$, as described in section 4.2. Further details on the selection of $\epsilon = 0.12$ can be found in Appendix E. We subsequently assess these results against subsets produced by the aforementioned post-hoc methods over standard-trained models (i.e., with only a prediction output, and no explanation) and examine their capacity to uphold robust sufficiency. Figure 3 illustrates visualizations of these results, while Table 1 provides a summary of the empirical evaluations.

Table 1: SST with robust masking vs. post-hoc methods. Results depict average explanation size (%), 10-minute timeouts (%), average time (seconds), and robust faithfulness (%).

|  | MNIST | | | | CIFAR-10 | | | | IMAGENET | | | |
|---|---|---|---|---|---|---|---|---|---|---|---|---|
|  | Size | T/O | Time | Faith. | Size | T/O | Time | Faith. | Size | T/O | Time | Faith. |
| Anchors | 8.98 | 0 | 0.11 | 97.51 | 20.88 | 0 | 0.8 | 86.47 | 20.29 | 1.09 | 118.6 | 88.94 |
| SIS | 2.35 | 0 | 4.34 | 98.08 | 0.87 | 0.14 | 195.25 | 84.64 | 0.15 | 0 | 262.42 | 77.32 |
| GS | 40.36 | 0 | 1.13 | 97.56 | 60.7 | 0 | 14.61 | 92.41 | 78.97 | 0 | 266.5 | 90.92 |
| **SST** | 1.42 | 0 | $\approx 10^{-6}$ | 99.28 | 12.99 | 0 | $\approx 10^{-5}$ | 90.43 | 0.46 | 0 | $\approx 10^{-4}$ | 80.88 |

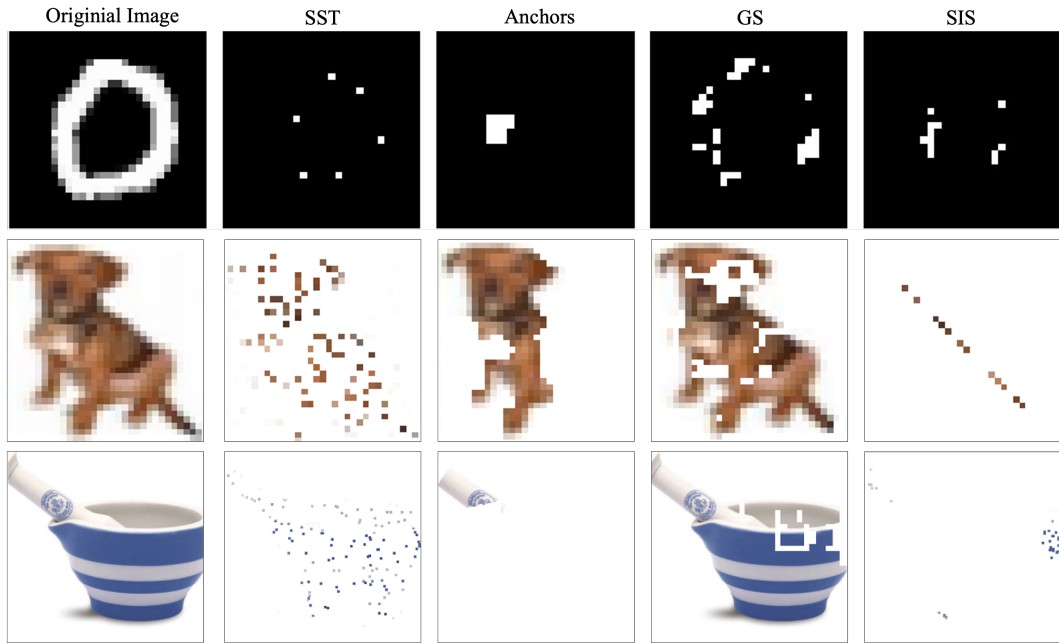

Figure 3: Examples of sufficient reasons produced by SST compared to the ones generated by post-hoc approaches for MNIST, CIFAR-10, and IMAGENET. Additional examples appear in appendix J.

Results in Table 1 indicate that across all configurations, SST generated explanations with substantially greater efficiency than post-hoc methods. SST typically produced smaller explanations, especially in all comparisons with GS and Anchors, and in MNIST against SIS. Additionally, SST explanations often demonstrated greater faithfulness, notably in MNIST against all post-hoc methods, and in CIFAR-10 compared to Anchors and SIS. Typically, SIS offered concise subsets but at the cost of low faithfulness, while GS provided faithful explanations that were considerably larger. Anchors yielded more consistent explanations, yet they were often either significantly larger or less faithful compared

to SST. Separately, concerning the accuracy shift for SST, MNIST showed a minor increase of $0.15\%$, whereas CIFAR-10 experienced a decrease of $1.96\%$, and IMAGENET saw a reduction of $4.7\%$. Of course, by reducing the impact of the cardinality and faithfulness loss coefficients, one can maintain accuracy on all benchmarks, but this may compromise the quality of the explanations.

**Different Masking Strategies for images.** We aim to analyze different masking strategies for SST in image classification, focusing on MNIST. We utilize two additional masking methods: *probabilistic* masking, where the assignments of features in the complement $\overline{S}$ are sampled from a uniform distribution to enhance probabilistic faithfulness, and *baseline* masking, where all the features in $\overline{S}$ are set to zero (using the trivial baseline $\mathbf{z} := \mathbf{0}_n$), hence optimizing baseline faithfulness. This specific baseline is intuitive in MNIST due to the inherently zeroed-out border of the images. We compare these approaches to the same post-hoc methods as before, evaluating them based on their baseline, probabilistic, or robust faithfulness, and present these results in Table 2.

Table 2: SST (different masks) vs. post-hoc methods: Mean explanation size (%), accuracy gain (%), mean inference time (seconds), and baseline, probabilistic, and robust faithfulness (%).

|  | Masking | Size | Acc. Gain | Time | | Faithfulness | |
|---|---|---|---|---|---|---|---|
|  |  |  |  |  | Robust | Probabilistic | Baseline |
| Anchors | — | 8.98 | — | 0.11 | 97.51 | 91.06 | 16.06 |
| SIS | — | 2.35 | — | 4.34 | 98.08 | 90.72 | 46.1 |
| GS | — | 40.36 | — | 1.13 | 97.56 | 94.11 | 16.72 |
| **SST** | **robust** | 1.42 | +0.15 | $\approx \mathbf{10^{-6}}$ | 99.28 | — | — |
|  | **baseline** | 23.69 | -1.26 | $\approx \mathbf{10^{-6}}$ | — | — | 96.52 |
|  | **probabilistic** | 0.65 | +0.36 | $\approx \mathbf{10^{-6}}$ | — | 99.11 | — |

Results in Table 2 demonstrate the impact of SST with various masking strategies. Each strategy improved compared to post-hoc methods in its respective faithfulness measure. Robust and probabilistic masking strategies produced very small subsets and even slightly *increased* accuracy. In contrast, the baseline masking strategy resulted in larger subsets and a larger accuracy drop. This likely occurs because setting the complement $\overline{S}$ to a fixed baseline strongly biases the model towards OOD instances, whereas uniform sampling or a bounded $\epsilon$-attack are less prone to this issue.

**Faithfulness-Cardinality Trade-off.** Optimizing models trained with SST involves a trade-off between two primary factors: (i) the size of the generated subset and (ii) the faithfulness of the model. For instance, a full set explanation ($S := \{1, \ldots, n\}$) scores low in terms of the optimization for small subset sizes but guarantees perfect faithfulness. Conversely, an empty set explanation ($S := \emptyset$) achieves the best score for subset size due to its minimal cardinality but generally exhibits poor faithfulness. In our experiment, we aim to illustrate this trade-off, which can be balanced by altering the cardinality loss coefficient $\xi$. Greater $\xi$ values emphasize the subset size loss component, leading to smaller subsets at the cost of their faithfulness. Conversely, smaller $\xi$ values reduce the component's impact, resulting in larger subsets with better faithfulness. Figure 4 provides a comparative analysis of this dynamic using the MNIST dataset with baseline masking. For greater $\xi$ values, the cardinality of the mask is maximal. As $\xi \to 0$, the

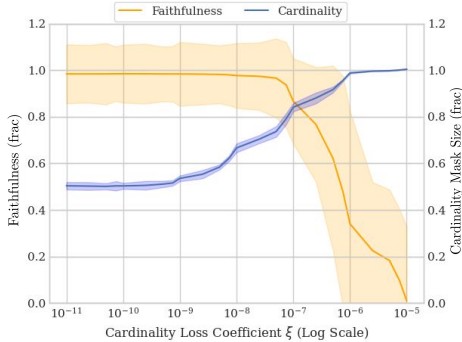

Figure 4: The faithfulness-cardinality tradeoff in baseline-masking SST models for MNIST with varying cardinality loss coefficients, $\xi$, shows that higher $\xi$ increases mask size $\overline{S}$ but reduces faithfulness, and vice versa.

explanation size converges to $50\%$ of the input. We believe this happens because the threshold is set to the default value of $\tau := 0.5$ (see Appendix E for more details), resulting in the arbitrary selection of a $50\%$ sufficient subset due to random initialization at the beginning of training. This subset is likely sufficient to determine the prediction on its own (as demonstrated in Table 2, even $23.69\%$ of

the input may be sufficient for a prediction). Consequently, as the cardinality loss coefficient gets very close to zero, the parameters of the explanation component (i.e., $h_2$, as discussed in Section 4) remain nearly unchanged throughout training, with the explanation size fixed at $50\%$.

## 5.2 SUFFICIENT SUBSET TRAINING FOR LANGUAGE

Table 3: Baseline/probabilistic SST (B-SST, P-SST) vs. post-hoc techniques: Mean inference time (seconds), 10-minute timeouts (%), mean explanation size (%), and probabilistic and baseline faithfulness (%)

| | IMDB | | | | | SNLI | | | | |
|---|---|---|---|---|---|---|---|---|---|---|
| | Time | T/O | Size | P-Faith. | B-Faith. | Time | T/O | Size | P-Faith. | B-Faith. |
| Anchors | 22.42 | 20.05 | 1.12 | 89.1 | 39.6 | 94.78 | 3.6 | 10 | 35.24 | 40.24 |
| S-Anchors | 19.65 | 12.26 | 1.16 | 89.14 | 23.37 | 160.92 | 9.76 | 18 | 28.55 | 33.94 |
| SIS | 14.9 | 0 | 22.57 | 88.97 | 97.89 | 23.13 | 0 | 12.74 | 44.84 | 49.29 |
| **B-SST** | $\approx 10^{-3}$ | 0 | 28.96 | — | **98.05** | $\approx 10^{-3}$ | 0 | 39 | — | **95.88** |
| **P-SST** | $\approx 10^{-3}$ | 0 | 49.7 | **95.67** | — | $\approx 10^{-3}$ | 0 | 53.69 | **95.35** | — |

We assess our results using two prevalent language classification tasks: IMDB sentiment analysis (Maas et al. (2011)) and SNLI (Bowman et al. (2015)). All models are trained over a BERT base (Devlin et al. (2018)). We implement two typical masking strategies for testing sufficiency in language classification (Hase et al. (2021); Vafa et al. (2021)): (i) using the pre-trained MASK token from BERT (Devlin et al. (2018)) as our baseline (baseline masking) and (ii) randomly sampling features uniformly (probabilistic masking) . We evaluate the performance of SST-generated subsets against post-hoc explanations provided by either Anchors or SIS. In the Anchors library, an additional method assesses sufficiency for text domains by sampling words similar to those masked, leading us to two distinct configurations which we denote as: *Anchors* and *Similarity-Anchors* (*S-Anchors*).

Table 3 shows that in the language setting, similarly to vision, SST obtains explanations much more efficiently than post-hoc methods. Although SST produces larger subsets than post-hoc methods, these subsets are significantly more faithful. For instance, SST achieves a baseline faithfulness of $98.05\%$ on IMDB, compared to only $23\%$ for S-Anchors. The convergence of SST towards larger subsets likely stems from our optimization of both faithfulness *and* minimal subset cardinality. Increasing the cardinality loss coefficient may yield smaller subsets but could decrease faithfulness. Separately, SST maintained comparable accuracy levels for both of these tasks — the decrease in accuracy for IMDB was $0.87\%$ and $0.89\%$ with probabilistic and baseline masking, respectively. For SNLI, probabilistic masking resulted in a $0.61\%$ drop, whereas baseline masking saw a smaller decrease of $0.32\%$.

**Baseline-Sufficiency (negative):**
I *found this* movie really hard …
wandering *off* the tv … *Don't* bother with it.

**Probabilistic-Sufficiency (negative):**
I *found* this movie *really hard* … *wandering*
*off the tv* …

Figure 5: Explanations generated by SST using baseline vs. probabilistic masking. When each token in the complement $\overline{S}$ is replaced with the MASK token, the prediction stays negative. In the probabilistic setting, the prediction remains negative when values from $\overline{S}$ are randomly sampled. Further examples can be found in Appendix J.

Explanations generated using probabilistic masking were larger on average compared to those generated using baseline masking. This is likely because random token perturbations present a more challenging constraint than fixing the MASK token. Figure 5 illustrates two explanations produced by SST using probabilistic or baseline masking for IMDB. Although both are intuitive, the baseline sufficient reason suffices by fixing only specific relevant terms like "don't" and "off". These terms are enough to preserve the negative classification when all other tokens are fixed to the MASK token. However, the probabilistic subset has larger negating phrases like "wandering off the TV" or "really hard" because it involves random perturbations of the complementary set, which may potentially shift the classification to positive.

## 6    RELATED WORK

**Post-hoc sufficient reason techniques.**    Some common post-hoc approaches include An-chors (Ribeiro et al. (2018)), or SIS (Carter et al. (2019)), and typically seek to obtain concise subsets which uphold probabilistic sufficiency (Wang et al. (2021a); Ribeiro et al. (2018); Blanc et al. (2021)) or baseline sufficiency (Hase et al. (2021); Chockler et al. (2021); DeYoung et al. (2019)). A different set of works is dedicated to obtaining robust sufficient reasons, where sufficiency is determined for some entire discrete or continuous domain. These are often termed *abductive explanations* (Ignatiev et al. (2019); Bassan et al. (2023)) or *prime implicants* (Ignatiev et al. (2019); Shih et al. (2018)). Due to the complexity of producing such explanations, they are commonly obtained on simpler model types, such as decision trees (Izza et al. (2020)), tree ensembles (Izza & Marques-Silva (2021); Ignatiev et al. (2022)), linear models (Marques-Silva et al. (2020)), or small-scale neural networks (Ignatiev et al. (2019); La Malfa et al. (2021); Bassan & Katz (2023)).

**Self-explaining neural networks.** Unlike post-hoc methods, some techniques modify training to improve explanations (Ismail et al. (2021); Chen et al. (2019b); Hase et al. (2021); Vafa et al. (2021); Yan & Wang (2023)) or develop *self-explaining* models that inherently provide interpretations for their decisions. Self-explaining models typically provide inherent *additive feature attributions*, assigning importance weights to individual or high-level features. For example,  Chen et al. (2019a) and  Wang & Wang (2021) describe model outputs by comparing them to relevant "prototypes", while Alvarez Melis & Jaakkola (2018) derive concept coefficients from feature transformations. Other approaches, like those by Agarwal et al. (2021) and Jain et al. (2020), focus on feature-specific neural networks or apply classifiers to snippets for NLP explanations, respectively.  Koh et al. (2020) predict outcomes based on high-level concepts. Due to their focus on *additive* attributions, these methods generally operate under the implicit or explicit assumption that the model's behavior can be approximated as *linear* under certain conditions (Yeh et al. (2019); Lundberg & Lee (2017)). This assumption is crucial as it underpins the approach of decomposing a model's output into independent, additive contributions from individual features, and may hence overlook nonlinear interactions among features (Slack et al. (2020); Ribeiro et al. (2018); Yeh et al. (2019)). In contrast, we propose a self-explaining framework that provides a distinct form of explanation — sufficient reasons for predictions — which do not depend on assumptions of underlying linearity (Ribeiro et al. (2018)). To our knowledge, this is the first self-explaining framework designed to generate such explanations.

## 7    LIMITATIONS

A primary limitation of SST is the need to tune additional hyperparameters due to multiple op-timization objectives. We simplified this by conducting a grid search solely over the cardinality loss coefficient $\xi$, as detailed in Appendix E. Additionally, the dual-propagation training phase is twice more computationally demanding than standard training, and this complexity escalates with challenging masking strategies, such as robust masking (see Appendix F for more details). However, this investment in training reduces the need for complex *post*-training computations, as highlighted in Section 5. Lastly, overly optimizing our interpretability constraints can sometimes reduce predictive accuracy. Our experiments showed this was true for IMAGENET, though all other benchmarks retained comparable accuracy levels (within $2\%$, and under $1\%$ for all language-model benchmarks). Interestingly, accuracy even improved under some configurations for MNIST classification. Future research endeavors could further investigate the conditions under which SST can be utilized not only for its inherent explanations but also to enhance predictive accuracy.

## 8    CONCLUSION

Obtaining (post-hoc) minimal sufficient reasons is a sought-after method of explainability. However, these explanations are challenging to compute in practice because of the computationally intensive processes and heavy dependence on OOD assignments. We further analyze these concerns and propose a framework to address them through *sufficient subset training* (*SST*). SST generates small sufficient reasons as an inherent output of neural networks with an optimization goal of producing subsets that are both *sufficient* and *concise*. Results indicate that, relative to post-hoc methods, SST provides explanations that are consistently smaller or more faithful (or both) across different models and domains, with significantly greater efficiency, while preserving comparable predictive accuracy.

## ACKNOWLEDGMENTS

This research was funded by the European Union's Horizon Research and Innovation Programme under grant agreement No. 101094905 (AI4GOV). We also extend our sincere thanks to Guy Amit, Liat Ein Dor, and Beat Buesser from IBM Research for their valuable discussions.

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

# Appendix

The following appendix is organized as follows:

**Appendix A** contains extended background on sufficient explanations and related work.
**Appendix B** contains background for the computational complexity proofs.
**Appendix C** contains the proof of theorem 1.
**Appendix D** contains the proof of theorem ii.
**Appendix E** contains technical specifications, related to the models, and training.
**Appendix F** contains information regarding the training time of SST compared to standard training.
**Appendix G** includes an experiment on the generalization of various masks across different sufficiency settings.
**Appendix H** includes an experiment on the extension of SST to simplified inputs.
**Appendix I** contains additional ablation results.
**Appendix J** contains supplementary results.

## A EXTENDED RELATED WORK AND BACKGROUND

**Sufficiency-based explanations.** Sufficiency-based explanations are a widely sought-after approach in explainability, with numerous methods proposed to achieve them (e.g., (Ribeiro et al., 2018; Carter et al., 2019; Ignatiev et al., 2019; Bassan & Katz, 2023)). The core idea is to identify a minimal subset of features that determine a prediction, allowing one to focus on the essential "reason" for the outcome while excluding non-relevant features. Notably, unlike the more prevalent additive attribution methods (Ribeiro et al., 2016; Lundberg & Lee, 2017; Sundararajan et al., 2017), these explanations do not depend on approximate linearity assumptions (Ribeiro et al., 2018; Yeh et al., 2019). From a human perspective, prior research has demonstrated their effectiveness; for instance, seeing multiple sufficiency-based explanations helps users predict model decisions more accurately compared to relying on additive attributions (Ribeiro et al., 2018).

The concept of sufficient explanations has been extensively explored in the literature. The general notion of a sufficient explanation is closely linked to ideas from actual causality (Halpern, 2016) and has been examined in several works (Chockler et al., 2021; Chockler & Halpern, 2024; Chockler et al., 2024; Kelly et al., 2023; Watson et al., 2021). Furthermore, the *robust* class of sufficient explanations studied in this work also connects to research in formal logic (Marques-Silva, 2023; Darwiche & Marquis, 2021). In the broader case where the $\epsilon_p$ perturbation is unbounded, such explanations are sometimes referred to as *abductive explanations* (Ignatiev et al., 2019; Ordyniak et al., 2023; Bassan et al., 2023). A related but distinct concept is that of a *prime implicant* (Shih et al., 2018; Ignatiev et al., 2019). Given the high computational complexity of deriving such explanations (Barceló et al., 2020), significant research has focused on identifying simplified models where explanation computation remains tractable (Cooper & Marques-Silva, 2023; Audemard et al., 2021). Examples of such models include decision trees (Izza et al., 2020; Huang et al., 2021; Bounia & Koriche, 2023), KNNs (Barceló et al., 2025), tree ensembles (Izza & Marques-Silva, 2021; Ignatiev et al., 2022; Audemard et al., 2022b;a; Boumazouza et al., 2021), linear models (Marques-Silva et al., 2020; Subercaseaux et al., 2024), monotonic models (Marques-Silva et al., 2021; El Harzli et al., 2023), and small-scale neural networks (Ignatiev et al., 2019; Wu et al., 2024b; La Malfa et al., 2021; Bassan & Katz, 2023). These explanations are often facilitated by neural network verifiers (Wang et al., 2021b; Wu et al., 2024a; Fel et al., 2023).

**Self-explaining models.** The concept of a self-explaining neural network (SENN) was first introduced by (Alvarez Melis & Jaakkola, 2018) and advocates for training architectures that inherently provide interpretations for their decisions (Lee et al., 2022; Shwartz et al., 2020; Rajagopal et al., 2021; Guyomard et al., 2022; Guo et al., 2023; Zhang et al., 2022; Ji et al., 2025). This idea is closely related to other research on learning human-understandable "concepts" (Koh et al., 2020; Blazek & Lin, 2021; Espinosa Zarlenga et al., 2022; Marconato et al., 2022; Kim et al., 2023) and representative training-data "prototypes"(Chen et al., 2019a; Keswani et al., 2022; Hong et al., 2023). Additionally, related work explores training interventions designed to improve feature selection capabilities (Lemhadri et al., 2021; Zhang et al., 2024; Jethani et al., 2021; Jiang et al., 2024; Yang et al., 2022; Yamada et al., 2020).

## B  COMPUTATIONAL COMPLEXITY BACKGROUND

In this section, we will introduce formal definitions and establish a computational complexity background for our proofs.

**Computational Complexity Classes.** We start by briefly introducing the complexity classes that are discussed in our work. It is assumed that readers have a basic knowledge of standard complexity classes which include polynomial time (PTIME) and nondeterministic polynomial time (NP and coNP). We also will mention the common complexity class within the second order of the polynomial hierarchy, $\Sigma_2^P$. This class encompasses problems solvable within NP, provided there is access to an oracle capable of solving co-NP problems in $O(1)$ time. It is generally accepted that NP and co-NP are subsets of $\Sigma_2^P$, though they are believed to be strict subsets (NP, co-NP $\subsetneq \Sigma_2^P$) (Arora & Barak (2009)). Furthermore, the paper discusses the complexity class *PP*, which describes the set of problems that can be solved by a *probabilistic* Turing machine with an error probability of less than $\frac{1}{2}$ for all underlying instances.

**Multi Layer Perceptrons**. Our hardness reductions are performed over neural networks with ReLU activation functions. We assume that our neural network is some function $f : \mathbb{R}^n \to \mathbb{R}^c$ where $n$ is the number of input features and $c$ is the number of classes. The classification of $f$ is chosen by taking $\arg\max_j f(\mathbf{x})_j$. For us to resolve cases where two or more distinct classes are assigned the same value, we assume that the set of $c$ classes is *ordered* ($i_1 \succ i_2 \ldots \succ i_c$), and choose the first maximal valued class. In other words, if there exist a set of ordered classes $i_1, \ldots, i_k \in [c]$ for which $i_1 \succ i_2 \succ \ldots \succ i_k$ and $f(\mathbf{x})_{i_1} = \ldots = f(\mathbf{x})_{i_k} = \arg\max_j f(\mathbf{x})_j$ then, without loss of generality, $i_1$ is chosen as the predicted class.

In the described MLP, the function $f$ is defined to output $f := g^{(t)}$. The first (input) layer $g^{(0)}$ is $\mathbf{x} \in \mathbb{R}^n$. The dimensions of both the biases and the weight matrices are represented by a sequence of positive integers $\{d_0, \ldots, d_t\}$. We focus on weights and biases that are *rational*, or in other words, $W^{(j)} \in \mathbb{Q}^{d_{j-1} \times d_j}$ and $b^{(j)} \in \mathbb{Q}^{d_j}$. These parameters were obtained during the training of $f$. From our previous definitions of the dimensions of $f$ it holds that $d_0 = n$ and $d_t = c$. We focus here on *ReLU* activation functions. In other words, $\sigma^{(i)}$ is defined by $\text{ReLU}(x) = \max(0, x)$.

## C  PROOF OF THEOREM 1

**Theorem 1.** *Given a neural network classifier $f$ with ReLU activations, and $\boldsymbol{x} \in \mathbb{R}^n$, obtaining a cardinally minimal sufficient reason for $\langle f, \boldsymbol{x} \rangle$ is (i) NP-Complete for baseline sufficient reasons (ii) $\Sigma_2^P$-Complete for robust sufficient reasons and (iii) $NP^{PP}$-Hard for probabilistic sufficient reasons.*

*Proof.* We follow computational complexity conventions (Barceló et al. (2020); Arenas et al. (2022)), focusing our analysis on the complexity of the *decision* problem which determines if a set of features constitutes a cardinally minimal sufficient reason. Specifically, the problem involves determining whether a subset of features $S \subseteq \{1, \ldots n\}$ qualifies as a sufficient reason while also satisfying $|S| \leq k$. This formulation is consistent with and extends the problems addressed in earlier research (Barceló et al. (2020); Arenas et al. (2022); Wäldchen et al. (2021)). The complexity problem is defined as follows:

---
**MSR (Minimum Sufficient Reason)**:
**Input**: A neural network $f$, and an input instance $\mathbf{x} \in \mathbb{R}^n$.
**Output**: *Yes* if there exists some $S \subseteq \{1, \ldots, n\}$ such that $S$ is a sufficient reason with respect to $\langle f, \mathbf{x} \rangle$, and *No* otherwise
---

We observe that earlier frameworks evaluating the complexity of the *MSR* query concentrated on cases where $\mathbf{x} \in \{0, 1\}^n$, a more straightforward scenario. Our focus extends to broader domains encompassing both discrete and continuous inputs and outputs. We are now set to define our queries, aligning with the definitions of sufficiency outlined in our work:

---

**P-MSR (Probabilistic Minimum Sufficient Reason)**:
**Input**: A neural network $f$, some distribution $\mathcal{D}$ over the features, $0 \leq \delta \leq 1$, and $\mathbf{x}$.
**Output**: *Yes* if there exists some $S \subseteq \{1, \ldots, n\}$ such that $S$ is a probabilistic sufficient reason with respect to $\langle f, \mathbf{x} \rangle$, $\mathcal{D}$ and $\delta$, and *No* otherwise

---

**R-MSR (Robust Minimum Sufficient Reason)**:
**Input**: A neural network $f$, an input instance $\mathbf{x} \in \mathbb{R}^n$, and (possibly) some $\epsilon > 0$.
**Output**: *Yes* if there exists some $S \subseteq \{1, \ldots, n\}$ such that $S$ is a robust sufficient reason with respect to $\langle f, \mathbf{x} \rangle$ (over the $\epsilon$-ball surrounding $\mathbf{x}$, or over an unbounded domain) and *No* otherwise

---

**B-MSR (Baseline Minimum Sufficient Reason)**:
**Input**: A neural network $f$, an input instance $\mathbf{x} \in \mathbb{R}^n$, and some baseline $\mathbf{z} \in \mathbb{R}^n$.
**Output**: *Yes* if there exists some $S \subseteq \{1, \ldots, n\}$ such that $S$ is a baseline sufficient reason with respect to $\langle f, \mathbf{x} \rangle$, and $\mathbf{z}$, and *No* otherwise

---

We will begin by presenting the complexity proof for the *R-MSR* query:

**Lemma 1.** *Solving the R-MSR query over a neural network classifier $f$, an input $\boldsymbol{x} \in \mathbb{R}^n$, and (possibly), some $\epsilon > 0$, where $f$ has either discrete or continuous input and output domains is $\Sigma_2^P$-Complete.*

*Proof.* Our proof is an extension of the one provided by the work of Barceló et al. (2020), who provided a similar result for multi-layer perceptrons that are restricted to binary inputs and outputs. We expand on this proof and show how it can hold for any possible input and output discrete or continuous domains.

**Membership.** In the binary instance, proving membership in $\Sigma_2^P$ can be done by guessing some subset of features $S$ and then using a coNP oracle for validating that this subset is indeed sufficient concerning $\langle f, \mathbf{x} \rangle$. When dealing with binary inputs, validating that a subset of features is in coNP is trivial since one can guess some assignment $\mathbf{z} \in \{0, 1\}^n$ and polynomially validate whether $f(\mathbf{x}_S; \mathbf{z}_{\bar{S}}) \neq f(\mathbf{x})$ holds.

When addressing continuous domains, the process of guessing $S$ remains consistent, but the method for confirming its sufficiency changes. This alteration arises because the guessed certificate $\mathbf{z}$ may not be polynomially bounded by the input size and could be unbounded. When guessing values in $\mathbb{R}$, it becomes crucial to consider a limit on their size, which ties back to the feasibility of approximating these values with sufficient precision.

To determine the complexity of obtaining cardinally minimal sufficient reasons, it is often helpful to first determine the complexity of a *relaxed* version of this query, which simply asks for the complexity of validating whether one specific given subset is a sufficient reason. We will use this as a first step in solving our unresolved inquiry for proving membership for continuous domains. The problem of validating whether one subset is sufficient is formalized as follows:

---

**CSR (Check Sufficient Reason)**:
**Input**: A neural network $f$, a subset $S \subseteq \{1, \ldots, n\}$, and an input instance $\mathbf{x} \in \mathbb{R}^n$, and possibly some $\epsilon > 0$.
**Output**: *Yes*, if $S$ is a sufficient reason with respect to $\langle f, \mathbf{x} \rangle$ (over some $\epsilon$ ball, or over an unbounded domain), and *No* otherwise

---

The process of obtaining a cardinally minimal sufficient reason (the *MSR* query) is computationally harder than validating whether one specific subset is sufficient (the *CSR* query). However, it is often helpful to determine the complexity of *CSR*, as a step for calculating the complexity of *MSR*.

We adopt the proof strategy from Sälzer & Lange (2021), which demonstrated the complexity that the complexity of the *NN-Reachability* problem is NP-Complete — an intricate extension of the initial proof in Katz et al. (2017). Intuitively, in *continuous* neural networks, rather than guessing *binary* inputs as our witnesses, we can guess the *activation status* of each ReLU constraint. This approach allows us to address the verification of neural network properties efficiently using linear

programming. We begin by outlining the *NN-Reachability* problem as defined by Sälzer & Lange (2021):

**Definition 1.** *Let $f : \mathbb{R}^n \to \mathbb{R}^m$ be a neural network. We define the specification $\phi$ as a property that is a conjunction of linear constraints $\phi_{in}$ on the input variables $\mathbf{X}$ and a set of linear constraints $\phi_{out}$ on the output variables $\mathbf{Y}$. In other words, $\phi := \phi_{in}(\mathbf{X}) \land \phi_{out}(\mathbf{Y})$.*

---

**NNReach (Neural Network Reachability)**:
**Input**: A neural network $f$, an input specification $\phi_{in}(\mathbf{x})$, and an output specification $\phi_{out}(\mathbf{y})$
**Output**: *Yes* if there exists some $\mathbf{x} \in \mathbb{R}^n$ and some $\mathbf{y} \in \mathbb{R}^m$ such that the specification $\phi$ holds, i.e., $\phi_{in}(\mathbf{x})$ is true and $\phi_{out}(\mathbf{y})$ is true, and *No* otherwise

---

**Lemma 2.** *Solving the CSR query for neural networks with continuous input and output domains can be polynomially reduced to the NNReach problem.*

*Proof.* We will begin by demonstrating the unbounded version (where no $\epsilon$ is provided as input), followed by an explanation of how we can extend this proof to a specific $\epsilon$-bounded domain. Given an instance $\langle f, S, \mathbf{x} \rangle$ we can construct an instance $\langle f, \phi_{in}, \phi_{out} \rangle$ for which we define $\phi_{in}$ as a conjunction of linear constraints: $\mathbf{X}_i = \mathbf{x}_i$ for each $i \in S$. We use $\mathbf{X}_i = \mathbf{x}_i$ to denote an assignment of the input variable $\mathbf{X}_i$ to the assignment $\mathbf{x}_i \in \mathbb{R}$. Assume that $f(\mathbf{x})$ is classified to some class $t$ (i.e., it holds that $f(\mathbf{x})_t \geq f(\mathbf{x})_i$ for all $i \neq t$). Then, we define $\phi_{out}$ as a *disjunction* of linear constraints of the form: $\mathbf{Y}_t < \mathbf{Y}_i$ for all $i \neq t$.

$S$ is a sufficient reason with respect to $\langle f, \mathbf{x} \rangle$ if and only if setting the features $\mathbf{X}$ to there corresponding values in $\mathbf{x}$ determines that the prediction $f(\mathbf{x})$ always stays $t$. Hence, $S$ is not a sufficient reason if and only if there exist some assignments $\mathbf{x} \in \mathbb{R}^n$ and an assignment $\mathbf{y} \in \mathbb{R}^m$ for which all features in $\mathbf{X}$ are set to their values in $\mathbf{x}$ and $f(\mathbf{x})$ is *not* classified to $t$, implying that for all $i \neq t$ it holds that $f(\mathbf{x})_t < f(\mathbf{x})_i$. This concludes the reduction. This reduction can be readily adapted for scenarios where sufficiency is defined within a *bounded* $\epsilon$ domain. This involves specifying that for all $i \in \overline{S}$, we include constraints such as: $\mathbf{X}_i \leq \mathbf{x}_i + \epsilon$ and $\mathbf{X}_i \geq \mathbf{x}_i - \epsilon$.

$\square$

**Finishing the Membership Proof.** Membership of *MSR* for continuous input domains can now be derived from the fact that we can guess some partial assignment to some subset $S \in \{1, \ldots n\}$ of size $k$. Then we can utilize Lemma 2 to incorporate a coNP-oracle which validates whether $S$ is sufficient concerning $\langle f, \mathbf{x} \rangle$ over the continuous domain $\mathbb{R}^n$.

**Hardness.** Prior work (Barceló et al. (2020)) demonstrated that *MSR* is $\Sigma_2^P$-hard for MLPs limited to *binary* inputs and outputs. To comprehend how this proof might be adapted to continuous domains, we will initially outline the key elements of the proof in the binary scenario. This involves a reduction from the *Shortest-Implicant-Core* problem, which is known to be $\Sigma_2^P$-Complete (Umans (2001)). We begin by defining an *implicant* for Boolean formulas:

**Definition 2.** *Let $\phi$ be a boolean formula. An implicant $C$ for $\phi$ is a partial assignment of the variables of $\phi$ such that any assignment to the remaining variables evaluates to true.*

The reduction also makes use in the following Lemma (whose proof appears in Barceló et al. (2020)):

**Lemma 3.** *Any boolean circuit $\phi$ can be encoded into an equivalent MLP over the binary domain $\{0, 1\}^n \to \{0, 1\}$ in polynomial time.*

We will now begin by introducing the reduction for *binary*-input-output MLPs from the *Shortest-Implicant-Core* problem (Barceló et al. (2020)). The *Shortest-Implicant-Core* problem is defined as follows:

---

**Shortest Implicant Core**:
**Input**: A formula in disjunctive normal form (DNF) $\phi := t_1 \lor t_2 \ldots \lor t_n$, an implicant $C$ of $\phi$, and an integer $k$.
**Output**: *Yes*, if there exists an implicant $C' \subseteq t_n$ of $\phi$ of size $k$, and *No* otherwise

---

Initially, we identify $X_c$ as the set of features not included in $t_n$. Subsequently, each variable $x_j \in X_c$ can be divided into $k + 1$ distinct variables $x_j^1, \ldots x_j^{k+1}$. For each $i \in \{1, \ldots, k + 1\}$, we can

construct a new formula $\phi^{(i)}$ by substituting every instance of the variable $x_j$ in $X_c$ with $x_j^i$. Finally, we can define $\phi'$ as the conjunction of all the $\phi^{(i)}$ formulas. In summary:

$$
\begin{aligned}
\phi^{(i)} &:= \phi[x_j \to x_j^i, \forall x_j \in X_c], \\
\phi' &:= \bigwedge_{i=1}^{k+1} \phi^{(i)}
\end{aligned}
\tag{7}
$$

The reduction then applies Lemma 3 to transform $\phi'$ into an MLP $f$ with binary inputs and outputs. It also constructs a vector $\mathbf{x}$, setting the value to 1 for all features in $t_n$ that are positive literals, and 0 for the rest. According to the reduction in Barceló et al. (2020), then $\langle \phi, C, k \rangle \in$ *Shortest-Implicant-Core* if and only if there exists a subset $S$ of size $k$ such that, when $S$ is fixed to its values in $\mathbf{x}$, then $f(\mathbf{x}_S; \mathbf{z}_{\bar{S}})$ consistently predicts 1, for any assignment $\mathbf{z} \in \{0, 1\}^n$. This implies that $\langle \phi, C, k \rangle \in$ *Shortest-Implicant-Core* if and only if $\langle f, \mathbf{x}, k \rangle \in$ MSR for MLPs with binary inputs and outputs, demonstrating that *MSR* for binary-input-output MLPs is $\Sigma_2^P$-Hard.

However, it is important to note that the previous lemma may not apply when the MLP in question does not exclusively handle binary inputs and outputs. We will now demonstrate how we can transform $f$ to another MLP $f'$, in polynomial time. We then will prove that when $f$ is defined such that $f : \{0, 1\}^n \to \{0, 1\}$ and $f'$ is defined such that $f' : \mathbb{R}^n \to \mathbb{R}^2$, a sufficient reason of size $k$ exists for $\langle f, \mathbf{x} \rangle$ if and only if one exists for $\langle f', \mathbf{x} \rangle$. This reduction will extend the reduction from the *Shortest-Implicant-Core* to *MSR* for MLPs with non-binary inputs and outputs, thereby providing a proof of $\Sigma_2^P$-hardness for non-binary inputs and output MLPs.

Typically, $f'$ will be developed based on $f$, but it will include several additional hidden layers and linear transformations. To begin, we will introduce several constructions that are crucial for this development. We will specifically demonstrate how these constructions can be implemented using only linear transformations or ReLU activations, allowing them to be integrated into our MLP framework. The initial construction we will discuss was proposed by Sälzer & Lange (2021):

$$
z := \mathrm{ReLU}(\frac{1}{2} - x) + \mathrm{ReLU}(x - \frac{1}{2}) - \frac{1}{2}
\tag{8}
$$

In Sälzer & Lange (2021), it is demonstrated that the construction is equivalent to:

$$
z = \begin{cases} -x & if \ x < \frac{1}{2} \\ x - 1 & otherwise \end{cases}
\tag{9}
$$

From this, it can be directly inferred that $z = 0$ if and only if $x = 0$ or $x = 1$, and $z \neq 0$ for any other values of $x \in \mathbb{R}$. This construction will prove useful in our reduction. We will also utilize a second construction, which is described as follows:

$$
z := \mathrm{ReLU}(x) + (-1) \cdot \mathrm{ReLU}(-x) = |x|
\tag{10}
$$

**The reduction.** We will now explain how to convert $f$ into the appropriate $f'$ to ensure the validity of our reduction. We begin with the foundational structure of $f$ as the skeleton for $f'$. Initially, we add extra neurons to the first hidden layer and link them to the input layer as follows. Each input feature $\mathbf{x}_i$ is processed through a series of computations, which, leveraging the two constructions previously mentioned, can be performed solely using ReLU activations. Specifically, for each feature $\mathbf{x}_i$, we compute:

$$
\begin{aligned}
\mathbf{x}_i' := \mathrm{ReLU}(\mathrm{ReLU}(\frac{1}{2} - \mathbf{x}_i) + \mathrm{ReLU}(\mathbf{x}_i - \frac{1}{2}) - \frac{1}{2}) + \\
(-1) \cdot \mathrm{ReLU}(-\mathrm{ReLU}(\frac{1}{2} - \mathbf{x}_i) - \mathrm{ReLU}(\mathbf{x}_i - \frac{1}{2}) + \frac{1}{2}) = \\
|\mathrm{ReLU}(\frac{1}{2} - \mathbf{x}_i) + \mathrm{ReLU}(\mathbf{x}_i - \frac{1}{2}) - \frac{1}{2}|
\end{aligned}
\tag{11}
$$

The existing structure of $f$ (and thus $f'$) features a *single* output neuron, which, when given an input from the domain $\mathbf{x} \in \{0,1\}^n$, returns a single *binary* value from $\{0,1\}$. Let us denote the value of this output neuron as $o_1$. To develop $f'$, we introduce additional hidden layers that connect sequentially after $o_1$. Initially, we link $o_1$ to two new output neurons, denoted $o_{1,1}$ and $o_{1,2}$. These neurons are configured as follows: $o_{1,2}$ is directly tied to $o_1$ with a linear transformation of value 1, effectively making $o_{1,2} = o_1$. For the other output, we establish the weights and biases through the following computation:

$$o_{1,1} := \mathrm{ReLU}(o_1) - 1 \tag{12}$$

It is evident that the following equivalence is maintained:

$$o_{1,1} = \begin{cases} 1 & if \ o_1 = 0 \\ 0 & o_1 = 1 \end{cases} \tag{13}$$

We will now outline the construction of the *output* layer of $f'$, which consists of two outputs: $o_{2,1}$ and $o_{2,2}$. Adhering to our formalization of MLP, we assume the following order: $o_{2_1} \succ o_{2,2}$, without loss of generality. This means that in cases where a specific assignment $\mathbf{z} \in \mathbb{R}^n$ is processed through $f'$ and results in $o_{2_1} = o_{2,2}$, then $f'$ is classified under class $o_{2,1}$. Conversely, $f'$ is classified as class $o_{2,2}$ only if the value at $o_{2,2}$ is *strictly* larger than that at $o_{2,1}$. The constructions for $o_{2,1}$ and $o_{2,2}$ are as follows:

$$o_{2,1} := \mathrm{ReLU}(o_{1,1}) + (-1) \cdot \mathrm{ReLU}(-o_{1,1}) + \sum_{i \in [m]} \mathbf{x}'_i,$$

$$o_{2,2} := \mathrm{ReLU}(o_{1,2}) + (-1) \cdot \mathrm{ReLU}(-o_{1,2}) + \sum_{i \in [m]} \mathbf{x}'_i + o_{2,1} \tag{14}$$

Which is equivalent to:

$$o_{2,1} := |o_{1,1}| + \sum_{i \in [m]} \mathbf{x}'_i,$$

$$o_{2,2} := |o_{1,2}| + \sum_{i \in [m]} \mathbf{x}'_i + o_{2,1} \tag{15}$$

**Reduction Proof.** We will now establish the correctness of the reduction. Once more, we will begin by establishing correctness for the *unbounded* domain (where no $\epsilon$ is provided as input), before outlining the necessary adjustments for the bounded domain. Initially, by design, we note that $\mathbf{x} \in \{0,1\}^n$ because the input vector, derived from the *Shortest-Implicant-Core*, contains only binary features. Following the specified construction of $\mathbf{x}$ where all positive features in $t_n$ within the formula $\phi'$ are set to 1, and all other features are set to 0, it results in $t_n$ evaluating to True. Consequently, $\phi'$ evaluates to true, thereby causing $f$ to evaluate to 1. This means that when $\mathbf{x}$ is processed through $f$, the single output value $o_1$ is determined to be 1. Since the connections from the input layer to the output $o_1$ in $f'$ remain unchanged from those in $f$, when $\mathbf{x}$ is input into $f'$, $o_1$ also evaluates to 1.

From the design of $f'$, we observe that $o_{1,1} = 0$ and $o_{1,2} = 1$. Given that $\mathbf{x}$ comprises only binary values, our earlier construction ensures that all $\mathbf{x}'_i$ values are equal to 0. Consequently, this results in: $o_{2,1} = 0$ and $o_{2,2} = 1$, leading to the classification of $f$ under class $o_{2,2}$.

Additionally, we observe that by definition, $\mathbf{x}'_i \geq 0$ for all $i$, which implies that it invariably holds that:

$$[o_{2,1} = |o_{1,1}| + \sum_{i \in [m]} \mathbf{x}'_i \geq 0] \quad \wedge \quad [o_{2,2} = |o_{1,2}| + \sum_{i \in [m]} \mathbf{x}'_i + o_{2,1} \geq 0] \tag{16}$$

Since the precedence $o_{2,1} \succ o_{2,2}$ is established, it follows that $f'$ is classified under class $o_{2,1}$ if and only if $o_{2,1} = 0$. This condition arises because if $o_{2,1} \neq 0$, then it necessarily means that $o_{2,2} > o_{2,1}$

resulting in $f'$ being classified under $o_{2,2}$. Conversely, if $o_{2,2} = 0$, then $o_{2,1} = o_{2,2}$, and thus $f'$ is classified under $o_{2,1}$ by virtue of $o_{2,1} \succ o_{2,2}$.

Assuming that $\langle \phi, C, k \rangle \in$ *Shortest-Implicant-Core*, as previously mentioned, it follows that there is a subset $S$ of features, each of size $k$, where fixing the features in $S$ to their values in $\mathbf{x}$ results in the output $o_1$ in $f'$ consistently remaining at 1, irrespective of how the features in $\bar{S}$ are set within the binary domain $\{0, 1\}^n$. However, we need to establish a more robust claim —— that this sufficiency extends to any values assigned to the features of $\bar{S}$ from the continuous domain $\mathbf{z} \in \mathbb{R}^n$. Specifically, we need to demonstrate that when the values of the features in $S$ are fixed as they are in $\mathbf{x}$, the prediction of $f'$ consistently remains at $o_{2,2}$, no matter what *real* values are assigned to the features in $\bar{S}$.

We have established that fixing the values of $S$ to those in $\mathbf{x}$ results in $o_1$ in $f'$ consistently equaling 1, irrespective of how the values of $\bar{S}$ are set within $\{0, 1\}^n$. Given that $\mathbf{x}'_i \neq 0$ for all $i$ under our conditions, it follows that both $o_{2,1}$ and $o_{2,2}$ are greater than 0. We have previously demonstrated that if $o_{2,1} > 0$, then $f'$ is classified as $o_{2,2}$. Therefore, for any potential assignment of the features in $\bar{S}$ within the domain $\{0, 1\}^n$, $f'$ is classified under $o_{2,2}$. Since the original prediction is $o_{2,2}$, this confirms that there is indeed a subset $S$ of size $k$, which, when fixed at their values, ensures that the prediction of $f'$ remains consistently at $o_{2,2}$.

Let us consider the scenario where the values for the complementary set $\bar{S}$ are sourced not exclusively from the $\{0, 1\}^n$ domain, meaning they are derived from $\mathbf{z} \in \mathbb{R}^n \setminus \{0, 1\}^n$. In more precise terms, there exists at least one feature $i$ within $\bar{S}$ such that for the assignment $\mathbf{z} \in \mathbb{R}^n$, $\mathbf{z}_i \neq 1$ and $\mathbf{z}_i \neq 0$. We must demonstrate that under these circumstances, the prediction for $f'$ still stabilizes at $o_{2,2}$. For this particular feature $\mathbf{z}_i$, it is confirmed that:

$$\mathrm{ReLU}(\frac{1}{2} - \mathbf{z}_i) + \mathrm{ReLU}(\mathbf{z}_i - \frac{1}{2}) - \frac{1}{2} \neq 0 \tag{17}$$

This also suggests that within the newly constructed hidden layers of $f'$, there are some $\mathbf{x}'_i$ values such that:

$$\mathbf{x}'_i = |\mathrm{ReLU}(\frac{1}{2} - \mathbf{z}_i) + \mathrm{ReLU}(\mathbf{z}_i - \frac{1}{2}) - \frac{1}{2}| > 0 \tag{18}$$

We now can conclude that:

$$o_{2,1} = |o_{1,1}| + \sum_{i \in [m]} \mathbf{x}'_i > 0 \tag{19}$$

Consequently, $f$ is classified under class $o_{2,2}$ even when the features in $\bar{S}$ are assigned values from the domain $\mathbb{R}^n \setminus \{0, 1\}^n$. In summary, this indicates that there exists a subset $S$ which consistently remains sufficient for $\langle f', \mathbf{x} \rangle$ when assessing sufficiency across any possible value in $\mathbb{R}^n$.

For the converse direction of the reduction, let us assume that $\langle \phi, C, k \rangle \notin$ *Shortest-Implicant-Core*. This implies that there is no subset $S$ of size $k$ or smaller such that $f$ consistently evaluates to class $o_1$ when the values of $\mathbf{z}$ come solely from the binary domain $\{0, 1\}^n$. This also means that for any subset $S$ of size $k$ or smaller, there exists some $\mathbf{z} \in \{0, 1\}^n$ where $f(\mathbf{x}_S; \mathbf{z}_{\bar{S}})$ is evaluated to 0 (resulting in a misclassification). Since $(\mathbf{x}_S; \mathbf{z}_{\bar{S}})$ is entirely binary, all hidden neurons $\mathbf{x}'_i$ evaluate to 0, leading to $o_{2,1} = 0$ and thus, $f'$ is misclassified to $o_{2,1}$. This overall demonstrates that there is no subset $S$ of size $k$ or less that is sufficient concerning $\langle f', \mathbf{x} \rangle$, when the sufficiency of $S$ is assessed relative to $\mathbb{R}^n$. This completes the reduction.

We now need to demonstrate how to adjust the following reduction when $\epsilon$ is an input parameter, with the sufficiency of $S$ restricted to an $\epsilon$-ball rather than being unbounded. The reduction can simply set $\epsilon > 1$. Since $\mathbf{x} \in \{0, 1\}^n$, the sufficiency of $S$ is then defined over a continuous $\epsilon$-ball that includes all inputs in $\{0, 1\}^n$. Let's refer to this domain as $\mathbb{F}$. In our reduction, we have shown that any contrastive assignment achievable in $f'$ must also be achieved over the domain $\{0, 1\}^n$. Thus, sufficiency for $S$ in the domain $\mathbb{R}^n \setminus \{0, 1\}^n$ is equivalent to sufficiency in $\mathbb{F} \setminus \{0, 1\}^n$, which implies that our reduction is applicable even in such a *bounded* domain.

$\square$

**Lemma 4.** *Solving the B-MSR query over a neural network classifier $f$ with either continuous or discrete input and output domains is NP-Complete.*

*Proof.* We begin by proving membership. Membership in NP is established because one can guess a subset of features $S$ and then verify whether $f(\mathbf{x}_S; \mathbf{x}'_{\bar{S}}) \neq f(\mathbf{x})$ and whether $|S| \leq k$.

**Hardness.** We demonstrate hardness by presenting a reduction from the classic *CNF-SAT* problem, which is known to be NP-Complete and defined as follows:

---

**CNF-SAT**:
**Input**: A formula in conjunctive normal form (CNF): $\phi$.
**Output**: *Yes*, if there exists an assignment to the $n$ literals of $\phi$ such that $\phi$ is evaluated to True, and *No* otherwise

---

We denote $\mathbf{1}_n$ as a vector of size $n$ containing only 1's, and $\mathbf{0}_n$ as a vector of size $n$ containing only 0's. Given some input $\phi$, we can first assign $\mathbf{1}_n$ to $\phi$ (setting all variables to True). If $\phi$ evaluates to True, then there exists a Truth assignment to $\phi$, so the reduction can construct some trivially correct encoding. We can hence assume that $\phi(\mathbf{1}_n) = 0$. We now denote the following:

$$\phi_2 := (x_1 \wedge x_2 \wedge \ldots x_n),$$
$$\phi' := \phi \vee \phi_2 \tag{20}$$

$\phi'$, is no longer a CNF, however, we can still use Lemma 3 to transform $\phi'$ into an MLP $f$ which behaves equivalently to $\phi'$ under the domain $\{0,1\}^n$. The reduction then finally constructs $\langle f, \mathbf{x} := \mathbf{1}_n, \mathbf{x}' := \mathbf{0}_n, k = n - 1 \rangle$.

We first note that $\phi_2(\mathbf{x}) = 1$, and hence both $\phi'(\mathbf{x}) = 1$ and $f(\mathbf{x}) = 1$. If $\langle \phi \rangle \in$ *CNF-SAT*, then there exists an assignment of variables that evaluates $\phi$ to True. This means there is some assignment in $\{0,1\}^n$ that evaluates $f$ to True. Since we have assumed that this is not the assignment $\mathbf{x} = \mathbf{1}_n$, it must be some assignment $\mathbf{x}' \in \{0,1\}^n \neq \mathbf{x}$. This implies that there exists an assignment of size $k < n$ that evaluates $\mathbf{1}_n$ to True. Consequently, there exists a subset $S$ where $|S| \leq k$ and $f(\mathbf{1}_S; \mathbf{0}_{\bar{S}}) = f(\mathbf{x}_S; \mathbf{x}'_{\bar{S}}) = 1 = f(\mathbf{1}_n)$.

For the second direction, we assume that $\langle \phi \rangle \notin$ *CNF-SAT*. This implies that no assignment evaluates $\phi$ to True. Consequently, there is no assignment with fewer than $n$ variables that evaluates $\phi_2 = (x_1 \wedge x_2 \wedge \ldots \wedge x_n)$ to True. Therefore, there is no assignment of size $k = n - 1$ or less that evaluates either $\phi$ or $\phi_2$ to True. From this, we conclude that there is no subset $S \subseteq [n]$ of size $k$ or less for which $f(\mathbf{x}_S; \mathbf{x}'_{\bar{S}}) \neq f(\mathbf{x}) = f(\mathbf{1}_n) = 1$, thus completing the reduction.

$\square$

**Lemma 5.** *Solving the P-MSR query on a neural network classifier $f$ is $NP^{PP}$-Hard.*

*Proof.* The reduction is derived by integrating the proof from Wäldchen et al. (2021) with Lemma 3. The work in (Wäldchen et al. (2021)) established that finding a cardinally minimal probabilistic sufficient reason for a CNF classifier, given a discrete uniform distribution over $\{0,1\}^n$, is $NP^{PP}$-Hard. Using Lemma 3, we can transform $\psi$ into an MLP $f$, applicable to either discrete or continuous domains. This allows us to construct $\langle f, \mathbf{x}' := \mathbf{x}, k' = k \rangle$ and define $\mathcal{D}$ as the uniform distribution over $\{0,1\}^n$. Given that $\mathbf{x}$ is an input for a CNF classifier, it directly follows that $\mathbf{x} \in \{0,1\}^n$. It hence follows that a cardinally minimal probabilistic sufficient reason for $\psi$ exists if and only if one exists for $\langle f, \mathbf{x} \rangle$ concerning the distribution $\mathcal{D}$, thereby completing the reduction.

$\square$

We observe that since the previous proof establishes *hardness* over the uniform distribution, this hardness persists under any distributional assumption that includes this distribution, such as independent distributions or more expressive ones, like the Markovian distributions examined by (Marzouk & de La Higuera), or the Hidden Markov modeled distributions examined by (Marzouk et al., 2025).

# D  PROOF OF THEOREM II

**Theorem 2.** *Given a neural network classifier $f$ with ReLU activations, and $\boldsymbol{x} \in \mathbb{R}^n$, (i) $\forall \epsilon > 0$ approximating cardinally minimal robust sufficient reasons with $n^{\frac{1}{2}-\epsilon}$ factor is $\Sigma_2^P$-Hard, (ii) $\forall \epsilon > 0$ it is NP-Hard to approximate cardinally minimal probabilistic or baseline sufficient reasons with factor $n^{\frac{1}{2}-\epsilon}$.*

*Proof.* We follow common conventions and when discussing the inapproximability of obtaining cardinally minimal sufficient reasons, refer to the *optimization*-version of these problems, or in other words obtaining a cardinally minimal sufficient reason, instead of asking whether there exists such a subset of size $k$ (which is the decision version). To avoid confusion, we will refer to these problems as *R-MSR\**, *B-MSR\** and *P-MSR\**. We start by proving the first complexity class:

**Lemma 6.** *Given a neural network classifier $f$ with ReLU activations, and $\boldsymbol{x} \in \mathbb{R}^n$, $\forall \epsilon > 0$ approximating cardinally minimal robust sufficient reasons with $n^{\frac{1}{2}-\epsilon}$ factor (i.e., solving the R-MSR\* query) is $\Sigma_2^P$-Hard*

*Proof.* We first note a known inapproximability result for the *Shortest-Implicant-Core* problem (Umans (1999)) which will be used to prove the inapproximability result for our case:

**Lemma 7.** *Given a DNF formula $\psi$, then for all $\epsilon > 0$, approximating the Shortest Implicant Core of $\psi$ to within factor $n^{1-\epsilon}$ is $\Sigma_2^P$-Hard.*

We acknowledge, however, that despite the difficulty of the *MSR* query being established through a reduction from the *Shortest Implicant Core* problem (as proven in Barceló et al. (2020) and discussed in Lemma 1), this approach cannot be directly utilized because the reduction is not necessarily *approximation preserving*. Specifically, the reduction discussed in Lemma 1, which follows Barceló et al. (2020), involves an MLP with an input space considerably larger than that of the boolean formula. If the input size of the boolean circuit is $n$, then the input size of the constructed MLP in the worst-case scenario will be $n \cdot (k+1)$.

Note that if $k = n$, the problem has a trivial solution. Therefore, we can assume the problem is addressed for cases where $n > k$, or equivalently $n \geq k+1$. Let us denote the optimal solution for the *MSR* query in MLPs as $\text{OPT}_{\text{MSR}}$ and the optimal solution for the shortest implicant core as $\text{OPT}_{\text{CORE}}$. Given that $k$ remains constant in both reductions, it follows that $\text{OPT}_{\text{MSR}} = \text{OPT}_{\text{CORE}}$. However, we still need to demonstrate that the same approximation ratios are applicable.

Let us suppose there exists some $\epsilon > 0$ such that an algorithm can solve the optimization problem of obtaining a cardinally minimal sufficient reason for MLPs with an approximation factor of $n^{\frac{1}{2}-\epsilon}$. We will denote the solution provided by this algorithm as $k'$. Consequently, we observe that:

$$
\begin{aligned}
k' \leq \text{OPT}_{\text{MSR}} \cdot (n \cdot (k+1))^{\frac{1}{2}-\epsilon} = \text{OPT}_{\text{CORE}} \cdot (n \cdot (k+1))^{\frac{1}{2}-\epsilon} \leq \\
\text{OPT}_{\text{CORE}} \cdot (n \cdot n)^{\frac{1}{2}-\epsilon} \leq \text{OPT}_{\text{CORE}} \cdot n^{1-2\cdot\epsilon} \leq \text{OPT}_{\text{CORE}} \cdot n^{1-\epsilon}
\end{aligned}
\tag{21}
$$

This results in an $n^{1-\epsilon}$ factor approximation algorithm for the shortest implicant core problem. Consequently, we conclude that approximating the optimization problem of obtaining robust cardinally minimal sufficient reasons is $\Sigma_2^P$-Hard to approximate within a factor of $n^{\frac{1}{2}-\epsilon}$.

$\square$

**Lemma 8.** *Given a neural network classifier $f$ with ReLU activations, and $\boldsymbol{x} \in \mathbb{R}^n$, $\forall \epsilon > 0$ approximating cardinally minimal probabilistic sufficient reasons with $n^{1-\epsilon}$ factor is NP-Hard*

This result can be extracted from the inapproximability results for obtaining cardinally minimal probabilistic sufficient reasons for CNF classifiers that were proven by Wäldchen et al. (2021). This result is as follows:

**Lemma 9.** *Given a CNF classifier, $\psi$, and some instance $\boldsymbol{x}$, for all $\epsilon > 0$ obtaining a probabilistic sufficient reason concerning $\langle \psi, \boldsymbol{x} \rangle$ under the uniform distribution defined over $\{0,1\}^n$, is NP-Hard to approximate within factor $n^{1-\epsilon}$.*

Using Lemma 3, we can replicate the process described under Lemma 5. We begin with $\psi$ and develop an MLP $f$, ensuring that a cardinally minimal sufficient reason applicable to $f$ is also valid

for $\psi$. This reduction is approximation preserving, as both $k' := k$ and $n' := n$, indicating that the same approximation ratio is preserved. Consequently, the hardness results established for CNF classifiers are equally applicable to any MLP with discrete or continuous input domains.

**Lemma 10.** *Given a neural network classifier $f$ with ReLU activations, and $\boldsymbol{x} \in \mathbb{R}^n$, $\forall \epsilon > 0$ approximating cardinally minimal baseline sufficient reasons with $n^{1-\epsilon}$ factor is NP-Hard*

We will perform an approximation preserving reduction from the *Max-Clique* problem, which is known to be hard to approximate. The problem is defined as follows:

---

**Max-Clique**:
**Input**: A graph $G := (V, E)$
**Output**: a *clique* in $G$ (i.e., a subset of vertices $C \subseteq V$ where each two vertices in $C$ are adjacent), which has maximal cardinality.

---

The following inapproximability result is known for the Max-Clique problem (Håstad (1999)):

**Lemma 11.** *Given a graph $G = (V, E)$, for any $\epsilon > 0$ it is NP-Hard to approximate the maximum clique of $G$ with approximation factor $n^{1-\epsilon}$.*

We now present an (approximation-preserving) reduction from the Max-Clique problem to *B-MSR*$^*$ for neural networks. Consider a graph $G = (V, E)$. We define a Boolean formula $\psi$ that corresponds to $G$. $\psi$ will have $|V|$ variables: $x_1, \ldots, x_V$ (each variable representing a vertex in the graph). We define $\psi$ as follows:

$$\psi := \bigwedge_{(u,v) \notin E} (\neg x_u \vee \neg x_v) \tag{22}$$

Using Lemma 3, we encode $\psi$ into an MLP $f$. We assert that a subset $C \subseteq V$ constitutes a maximal clique if and only if $S := E \setminus C$ is a cardinally minimal baseline sufficient reason for $\langle f, \boldsymbol{0}_n \rangle$ with respect to the baseline $\boldsymbol{1}_n$.

First, we note that $f(\boldsymbol{0}_n) = 1$, since all variables are set to True in this case. Therefore, a sufficient reason for $f$ concerning the baseline $\boldsymbol{1}_n$ would be a subset of features $S$ such that setting the features in $\overline{S}$ to 1 keeps the classification as 1. We will prove that $C$ is a clique in $G$ if and only if $S = E \setminus C$ is a sufficient reason for $\langle f, \boldsymbol{0}_n \rangle$ concerning the baseline $\boldsymbol{1}_n$.

If $C$ is *not* a clique in $G$, there exist vertices $u, v \in C$ such that $(u, v) \notin E$. Therefore, the clause $\neg x_u \vee \neg x_v$ is included in $\psi$. Given that $u, v \in C$, they also belong to $\overline{S}$, and their features are modified from a 0 assignment to a 1 assignment. As a result, $\neg x_u \vee \neg x_v = $ False, leading to $\psi = $ False and $f(\boldsymbol{0}_S; \boldsymbol{1}_{\bar{S}}) = 0$. This demonstrates that in this case, $S$ is not sufficient concerning the baseline $\boldsymbol{1}_n$. Conversely, if $C$ is a clique in $G$, then for any $(u, v) \in C$, $(u, v) \in E$. This ensures that for any clause $\neg x_u \vee \neg x_v$ in $\psi$, both $x_u$ and $x_v$ remain fixed at the value 0, resulting in $\neg x_u \vee \neg x_v = $ True and thus $f(\boldsymbol{0}_S; \boldsymbol{1}_{\bar{S}}) = 1$. This confirms that $S$ is a sufficient reason concerning the baseline $\boldsymbol{1}_n$.

From the previous claim, it follows directly that a sufficient reason $S$ is of minimal cardinality if and only if the cardinality of $E \setminus C$ is minimal (when $C$ is a clique in $G$). This is equivalent to requiring that $C$ has maximal cardinality, thereby concluding our proof.

$\square$

# E  TECHNICAL SPECIFICATIONS

We provide detailed model specifications for reproducibility purposes. While our evaluations are conducted on specific benchmarks, the SST methodology can be adapted to any neural network classifier. We compare the sufficient reasons derived from SST-trained models with explanations from post-hoc methods on traditionally trained models. To ensure a fair comparison, we perform a separate grid search for each configuration to determine the optimal model. For traditional models, we optimize based solely on validation predictive accuracy, whereas for SST models, we consider a combination of accuracy, faithfulness, and subset cardinality. For SST-based models, we consistently

set the threshold value $\tau := \frac{1}{2}$ and the faithfulness coefficient $\lambda := 1$, conducting the grid search focusing only on the learning rate $\alpha$ and the cardinality coefficient $\xi$.

We begin by outlining general training details applicable to either all image or all language classification tasks, followed by a discussion of benchmark-specific implementations.

### E.1  IMAGE CLASSIFICATION EXPERIMENTS

All image classification configurations underwent a grid search across various learning rates $\alpha$, with values $\{10^{-2}, 10^{-3}, 10^{-4}, 10^{-5}, 10^{-6}, 10^{-7}\}$ for both standard and SST training. For SST, additional grid searches were conducted for the cardinality coefficient $\xi$ options: $\{10^{-5}, 10^{-6}, 10^{-7}, 10^{-8}, 10^{-9}, 10^{-10}, 10^{-11}\}$. All models were trained using the Adam optimizer, a batch size of 64, and held-out validation and test sets. For robust masking, a PGD $\ell_\infty$ attack with $\epsilon := 0.12$ was used, consisting of 10 steps each of size $\alpha' := 10^{-2}$. The value of $\epsilon := 0.12$ was selected to strike a balance between allowing a sufficiently large perturbation for learning, while avoiding excessive distortion of the image. On one hand, this creates a challenging task for learning (as opposed to smaller perturbations). On the other hand, it prevents excessive distortion (as might occur with larger perturbations), which could result in the complementary set $\overline{S}$ containing images that are entirely outside the desired distribution. Lastly, we apply a black background to mask the MNIST images, both for SST and post-hoc approaches, leveraging their naturally zeroed-out borders, which provides a clearer interpretation of the portions of the digit that fall within the sufficient reason. Moreover, for CIFAR-10 and IMAGENET images, we select images from these benchmarks where a specific object is highlighted against a white background. The sufficient reasons for these images are depicted over the same white background, allowing the explanation to remain focused on the object itself. This approach is applied consistently for both SST and the post-hoc methods.

### E.2  LANGUAGE CLASSIFICATION EXPERIMENTS

All language classification experiments utilized a pre-trained Bert-base (Devlin et al. (2018)) model (applicable to both standard training and SST). The grid search focused on learning rates $\alpha := \{2e^{-5}, 3e^{-5}, 5e^{-5}\}$, which are the typical values used for optimizing pre-trained Bert models (Devlin et al. (2018)). For SST, an additional grid search was conducted for the cardinality coefficient $\xi$ options: $\{10^{-4}, 10^{-5}, 10^{-6}, 10^{-7}, 10^{-8}\}$. Optimization was performed using the standard AdamW optimizer, with a batch size of 32 and held-out validation and test sets.

### E.3  MNIST

We trained a simple feed-forward neural network consisting of two hidden layers with sizes 128 and 64, respectively. The experiments were conducted using four Intel(R) Xeon(R) Gold 6258R @ 2.70GHz CPUs. For the standard training scenario, the optimal configuration selected was $\alpha := 10^{-4}$. For the SST-based models, the optimal configurations were: for robust masking $\alpha := 10^{-4}$ and $\xi := 10^{-7}$, and for both probabilistic masking and baseline masking, $\alpha := 10^{-3}$ and $\xi := 10^{-7}$.

### E.4  CIFAR-10

We train a ResNet18 architecture (He et al. (2016)) (which is *not* pre-trained) as our base model using four Intel(R) Xeon(R) Gold 6258R @ 2.70GHz CPUs and one Nvidia A100-SXM4-80GB GPU. We use a batch size of 64 and the Adam optimizer. For the robust masking-based SST configuration, the chosen hyperparameters are $\alpha := 10^{-3}$ and $\xi := 10^{-8}$.

### E.5  IMAGENET

We train an SST-based model on a *pre-trained* ResNet50 (He et al. (2016)) for IMAGENET classification using nine Intel(R) Xeon(R) Gold 6258R @ 2.70GHz CPUs and one Nvidia A100-SXM4-80GB GPU. The optimal configuration for robust masking was $\alpha := 10^{-6}$ and $\xi := 10^{-9}$.

### E.6 SNLI

We train our models using a pre-trained Bert-base (Devlin et al. (2018)) on the SNLI classification task. The training is conducted on 16 Intel(R) Xeon(R) Gold 6258R CPUs at 2.70GHz and one Nvidia A100-SXM4-80GB GPU. For standard training on SNLI, the optimal learning rate was set at $\alpha := 2e^{-5}$. In the SST scenario, the best configuration with probabilistic masking was $\alpha := 2e^{-5}$ and $\xi := 10^{-7}$, whereas, for baseline masking, it was $\alpha := 2e^{-5}$ and $\xi := 10^{-6}$.

### E.7 IMDB

Our models were trained using a Bert-base (Devlin et al. (2018)) pre-trained on IMDB sentiment analysis. The setup included 16 Intel(R) Xeon(R) Gold 6258R CPUs at 2.70GHz and one Nvidia A100-SXM4-80GB GPU. The optimal setting for standard training on IMDB was established at $\alpha := 2e^{-5}$. For SST, the most effective configuration with probabilistic masking was $\alpha := 2e^{-5}$ and $\xi := 10^{-7}$, while for baseline masking, it was $\alpha := 2e^{-5}$ and $\xi := 10^{-6}$.

## F TRAINING TIME

One drawback of our approach is its higher computational demand compared to conventional training. Specifically, the dual-propagation procedure we employ doubles the training cost. This increase is more pronounced with complex masking techniques like *robust* masking, which involves an adversarial attack at each iteration, slowing down the process and potentially complicating convergence. These issues are similar to those encountered in adversarial training (Shafahi et al. (2019)). However, we argue that this additional training effort reduces the necessity for computationally intensive *post*-computations to derive concise sufficient reasons, as highlighted in our paper. Furthermore, future studies could investigate various strategies to improve the efficiency of our training method, drawing on techniques used in adversarial training (Shafahi et al. (2019)).

We now will highlight the training time gain incorporated by SST for each particular benchmark, as opposed to a standard training configuration.

### F.1 IMAGE CLASSIFICATION TASKS

For MNIST, Standard training required 311.03 seconds (over 43 epochs), whereas baseline-masking SST was completed in just 26.55 seconds (over 2 epochs). This significant speed increase in the training, despite involving a dual-propagation process, was primarily due to the optimal learning rate for SST with baseline masking being set at $10^{-3}$, compared to $10^{-4}$ for standard training. For probabilistic masking, the training duration was 180.83 seconds (over 11 epochs), where the enhanced efficiency stemmed, again, from achieving optimal results at a higher learning rate. However, robust masking, which is more computationally demanding, took substantially longer, clocking in at 1796.99 seconds (over 49 epochs). For IMAGENET, standard training using the robust masking configuration ran for 287056.81 seconds (over 22 epochs) while the parallel standard-training configuration for IMAGENET ran for 74141.85 seconds, over 43 epochs (around 4 times faster). For CIFAR-10, SST using the robust masking that was mentioned in the paper took 3159.15 seconds (over 53 epochs) compared to standard training which took 477.42 seconds (over 49 epochs).

### F.2 LANGUAGE CLASSIFICATION TASKS

For IMDB sentiment analysis, standard training ran for 4974.33 seconds (over 1 epoch), while SST with baseline masking ran for 9713.42 seconds (over 2 epochs) and SST with probabilistic masking clocked at 21516.37 seconds (over 3 epochs). For SNLI classification, standard training ran for 5353.71 seconds (over 2 epochs), while SST with a probabilistic masking ran for 6084.66 seconds (over 1 epoch), SST with a baseline masking ran for 10440 seconds (over 2 epochs).

## G GENERALIZATION BETWEEN SUFFICIENCY SETTINGS

In this section, we examine SST's ability to generalize across different types of sufficiency configurations. Our work considers various sufficiency forms, including baseline, probabilistic, and robust

masking configurations. Users can select the masking procedure that aligns with their preferred sufficiency criterion. Stricter sufficiency constraints typically lead the model to select larger subsets, while looser constraints result in smaller ones. However, training SST with a specific masking configuration for one form of sufficiency does not necessarily ensure strong generalization to other forms. A potential approach to improve generalization is to train SST with multiple masking configurations, such as varying the masking setting across batches — an avenue worth exploring in future work. Nevertheless, even when trained with a single masking configuration, there are cases where one sufficiency form may implicitly imply another, particularly when it encompasses other forms. For instance, uniform sampling within a bounded perturbation may be naturally subsumed by a robust masking strategy applied to the same perturbation.

We conducted an additional experiment in which we trained each model from the experiments in Tables 2 and 3 (covering MNIST, SNLI, and IMDB) to evaluate the impact of different masking configurations on various forms of sufficiency. The results are presented in Table 4.

Table 4: The generalization of various masking configurations to different sufficiency conditions.

| | Masking | Robust | Faithfulness Probabilistic | Baseline |
|---|---|---|---|---|
| **MNIST** | **robust** | 99.28 | 99.32 | 11.82 |
| | **baseline** | 98.91 | 98.38 | 96.52 |
| | **probabilistic** | 98.85 | 99.11 | 8.16 |
| **SNLI** | **baseline** | — | 44.81 | 95.88 |
| | **probabilistic** | — | 95.35 | 93.12 |
| **IMDB** | **baseline** | — | 75.7 | 98.05 |
| | **probabilistic** | — | 95.67 | 77.7 |

Table 4 presents results highlighting key generalization patterns across different tasks. For language tasks, we observe the following: In SNLI, probabilistic masking demonstrates high baseline faithfulness, whereas baseline masking exhibits considerably lower probabilistic faithfulness, suggesting superior generalization under probabilistic constraints. In IMDB, both probabilistic and baseline masking achieve moderate faithfulness in the opposite masking configuration. In the MNIST experiment, a more intricate dynamic emerges. Robust masking and probabilistic sampling remain within a bounded $\epsilon_p$ domain, while the baseline configuration is significantly out-of-distribution (OOD), making the baseline task more challenging. This is reflected in the larger subset size (23.69%) produced by SST for the baseline compared to the robust and probabilistic settings. As a result, the larger sufficient reason derived from the baseline generalizes well across other configurations, maintaining high faithfulness. Conversely, subsets from probabilistic and robust masking generalize effectively to each other but perform poorly on the baseline, likely due to the baseline's pronounced OOD nature. These findings emphasize the complex interaction between domain selection, sampling distribution, and baseline properties in influencing generalization. We believe further exploration into the generalization capabilities of SST across masks presents an exciting research direction.

### G.1 DISCUSSION ON VARIATIONS OF THE FAITHFULNESS METRIC

Similar to the challenges that arise when generalizing across different sufficiency configurations, a similar concern extends to the *evaluation* of explanations — specifically, the computation of faithfulness. As with many explanation frameworks, defining a sufficient reason can be ambiguous, particularly due to the varying interpretations of "missingness" in the complement. Our work addresses this inherent limitation, which is prevalent in many post-hoc methods, by evaluating and training explanations across diverse sufficiency formulations. However, we recognize that no single definition can fully encapsulate a model's internal behavior, and this limitation also applies to faithfulness metrics assessing the sufficiency of subsets. This issue is widespread in many XAI methods, including SHAP (Sundararajan & Najmi, 2020), and metrics such as infidelity (Yeh et al., 2019), where different treatments of "missing" features lead to variations in explanation definitions and evaluation metrics. By exploring diverse sufficiency criteria, our work contributes to a more comprehensive understanding of this class of explanations.

## H    EXTENSIONS TO SIMPLIFIED INPUT SPACES

It is evident that SST can be trained using simplified input spaces instead of individual features such as pixels or tokens. These simplified spaces — such as super-pixels, sentences, or other meaningful units — can, in some cases, enhance the human interpretability of the resulting explanations, as explored in previous works (e.g., (Ribeiro et al., 2016; 2018)). To evaluate this approach, we conducted an experiment on MNIST and CIFAR-10, using a straightforward simplified input setting of $2 \times 2$ pixel patches. Accordingly, we adjusted the output vector to match the number of super-pixels and applied masking at the level of these simplified inputs.

For consistency, we used the same hyperparameter choices as in the original (non-simplified input space) grid search. However, a dedicated grid search for this specific configuration could yield improved results. Our findings indicate that, for MNIST, the average explanation size was 4.26%, with 90.53% faithfulness and a minimal accuracy drop of -0.34%. In contrast, for CIFAR-10, we observed a -4.54% accuracy drop, 91.74% faithfulness, and an explanation size of 22.93%. These results suggest that, under this particular setting, the original (non-simplified) input space produced better overall performance metrics. However, from a human interpretability standpoint, simplified inputs can provide more intuitive and comprehensible explanations. We include examples of these simplified explanations alongside their regular counterparts in Figure 6.

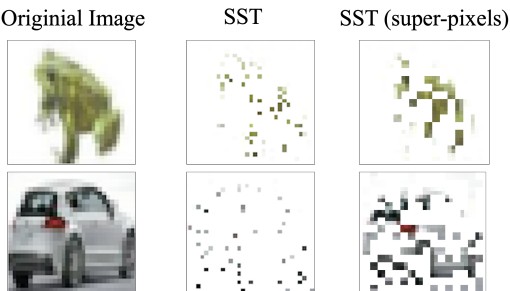

Figure 6: Examples of comparisons between explanations generated by SST using superpixels with a $2 \times 2$ patch versus those based on individual pixels.

## I    ADDITIONAL ABLATION EXPERIMENTS

In this section, we provide additional ablation results for our experiments. Specifically, we examine the effects of varying both the $\epsilon_p$ perturbation and the $\alpha$ step size in the robust masking experiments, as well as adjusting the $\tau$ selection threshold parameter. The results are summarized in Tables 5 to 7, with a detailed discussion of each experiment provided separately.

### I.1    VARYING THE $\epsilon_p$-BOUND IN ROBUST MASKING

We perform an ablation study by varying the $\epsilon_p$ attack in robust masking on CIFAR-10, using the same configurations as in our main experiment. In general, increasing $\epsilon_p$ makes it more challenging to satisfy the sufficiency conditions of the explanation. Consequently, users can select different values based on the desired "degree" of sufficiency. A higher $\epsilon_p$ can lead to larger explanation sizes and/or reduced faithfulness, or it may result in decreased model accuracy. The specific metric affected depends on the relative weighting of different loss coefficients during training, which influence the optimization objectives.

As shown in Table 5, increasing $\epsilon_p$ within our configurations generally leads to lower faithfulness and larger explanation sizes. However, we note that excessively large values of $\epsilon_p$ can reduce the effectiveness of gradient perturbations, a well-known issue in adversarial training (Addepalli et al., 2022). Thus, it is not surprising that at very high perturbations (e.g., above $0.25$), model accuracy deteriorates significantly. Addressing the challenges posed by such large perturbations, which is also relevant to adversarial training, remains an open problem for future research.

Table 5: An ablation study examining the effect of varying $\epsilon_p$ perturbations in robust masking on the CIFAR-10 benchmark.

| $\epsilon_p$ (%) | Acc. (%) | Faith. (%) | Size (%) |
|---|---|---|---|
| 0.01 | -0.13 | 99.31 | 10.2 |
| 0.05 | +0.7 | 95.09 | 0.91 |
| 0.1 | -1.58 | 91.46 | 9.78 |
| 0.12 | -1.96 | 90.43 | 12.99 |
| 0.15 | -1.45 | 88.03 | 11.91 |
| 0.2 | -3.32 | 85.34 | 25.49 |
| 0.25 | -4.58 | 79.9 | 17.48 |
| 0.3 | -9.92 | 78.88 | 35.9 |
| 0.35 | -5.97 | 70.94 | 19.2 |
| 0.4 | -10.25 | 64.95 | 34.53 |
| 0.45 | -14.68 | 64.95 | 34.53 |
| 0.5 | -14.09 | 68.05 | 31.99 |

Table 6: An ablation study examining the effect of varying $\alpha$: the step size in robust masking (with a fixed $\epsilon_p := 0.12$ perturbation) on the CIFAR-10 benchmark.

| $\alpha$ (%) | Acc. (%) | Faith. (%) | Size (%) |
|---|---|---|---|
| 1/255 | -0.66 | 89.47 | 12.01 |
| 2/255 | -2.22 | 89.9 | 19.7 |
| 3/255 | -1.13 | 89.74 | 10.93 |
| 4/255 | -2.12 | 90.83 | 17.28 |
| 5/255 | -1.96 | 90.43 | 12.99 |
| 6/255 | -0.08 | 89.25 | 7.2 |
| 7/255 | -1.32 | 90.21 | 9.78 |
| 8/255 | -1.91 | 90.44 | 15.09 |
| 9/255 | -1.68 | 90.27 | 12.89 |
| 10/255 | -2.71 | 89.15 | 17.27 |

### I.2 Varying the step size $\alpha$ in robust masking.

We conduct an ablation study by adjusting the step size $\alpha$ in adversarial training on CIFAR-10, while keeping the adversarial perturbation $\epsilon_p$ fixed at $0.12$. The results are presented in Table 6. Similar to adversarial training, smaller values of $\alpha$ yield adversarial examples that are closer to the original input, whereas larger values generate more distant examples. Importantly, the faithfulness measure we present quantifies the proportion of test instances that remain sufficient under a robust mask for each configuration, all computed using the same attack strategy (i.e., with the same $\alpha$ value). These findings demonstrate that our model can adapt to different $\alpha$ values, producing relatively compact and faithful explanations. However, a substantial increase in step size leads to a decline in accuracy.

Table 7: An ablation study examining the effect of varying the selection threshold $\tau$ in probabilistic masking on the CIFAR-10 benchmark.

| $\tau$ (0-1 range) | Acc. Gain (%) | Faith. (%) | Size (%) |
|---|---|---|---|
| 0.1 | +0.39 | 73.88 | 3.96 |
| 0.2 | -0.61 | 81.22 | 5.81 |
| 0.3 | -0.23 | 81.26 | 2.55 |
| 0.4 | -1.57 | 84.68 | 9.80 |
| 0.5 | -0.09 | 87.54 | 9.08 |
| 0.6 | -1.62 | 88.32 | 12.26 |
| 0.7 | -1.79 | 96.44 | 11.80 |
| 0.8 | -2.08 | 86.97 | 12.63 |
| 0.9 | -1.24 | 86.43 | 6.65 |

### I.3 Varying the selection threshold $\tau$

Lastly, we performed an ablation study on different values of $\tau$ trained with probabilistic masking on CIFAR-10. The results are presented in Table 7. Overall, as observed in the results, our method adapts to varying values of $\tau$ by optimizing the model to learn different weights within the explanation component output vector. However, excessively high or low $\tau$ values tend to be more sensitive and may negatively impact certain key metrics. For instance, a low setting of $\tau := 0.1$ resulted in reduced faithfulness ($73.88\%$), while a high setting of $\tau := 0.8$ led to lower accuracy ($-2.08\%$). In contrast, setting $\tau := 0.5$ yielded comparable accuracy ($-0.09\%$).

## J Supplementary Results

In this section, we will present additional results to support our evaluations.

## J.1 MNIST

We begin by offering more image samples to contrast post-hoc explanations generated over MNIST with SST explanations trained using robust masking, for comparison purposes.

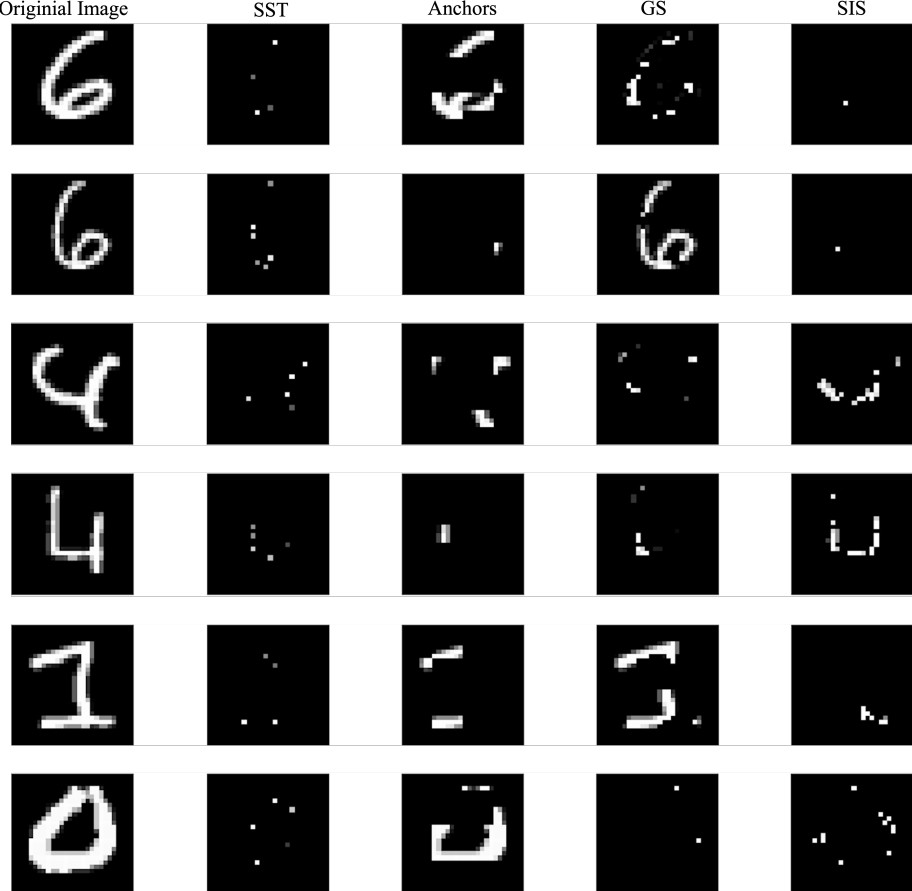

Figure 7: Examples of comparisons between explanations produced by *SST* compared to post-hoc approaches for MNIST

## J.2 CIFAR-10

We now transition to showcasing further comparison findings for CIFAR-10, contrasting post-hoc explanations with those generated using SST and robust masking techniques.

To further illustrate examples of explanations under various masking configurations, we conducted an ablation study using SST-based explanations on CIFAR-10. We implemented baseline masking by zeroing out the complement $\overline{S}$ (the trivial baseline: $\mathbf{z} := \mathbf{0}n$), *probabilistic* masking, where we sampled features over $\overline{S}$ from a uniform distribution either over the entire input domain ($\epsilon := 1$) or within a bounded region surrounding $\mathbf{x}$ ($\epsilon := 0.12$). Additionally, we performed robust masking by executing a PGD $\ell\infty$ attack within an $\epsilon := 0.12$ ball with 10 steps each of size $\alpha' := 10^{-2}$.

Regarding average explanation sizes: robust masking accounted for $12.99\%$, probabilistic masking with $\epsilon = 0.12$ was $9.40\%$, probabilistic masking with $\epsilon := 1$ was $50.75\%$, and baseline masking was $50.8\%$. The average faithfulness was $89.77\%$ for baseline masking, $87.92\%$ for bounded-probabilistic masking, $90.43\%$ for robust masking, and only $15.95\%$ for unbounded-probabilistic masking. Results are presented in Figure 9.

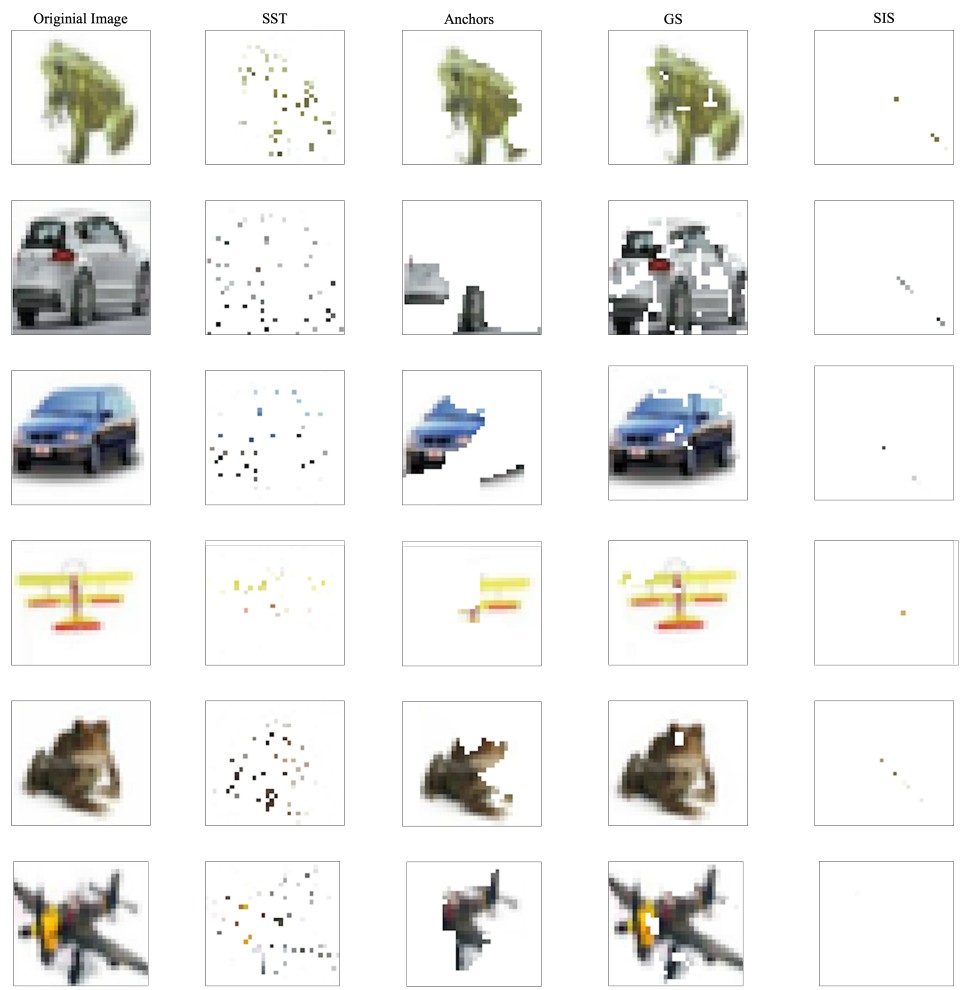

Figure 8: Examples of comparisons between explanations produced by *SST* compared to post-hoc approaches for CIFAR-10

### J.3 IMAGENET

Lastly, we present a comparative analysis of SST-based models and post-hoc approaches for IM-AGENET. Since IMAGENET inputs are high-dimensional and individual pixels are too small to be discernible to the human eye, we patch each pixel with the surrounding $5 \times 5$ pixels, centering the corresponding pixel for visualization purposes. This approach is applied in all visualizations of IMAGENET image results, including both SST and post-hoc methods.

In this segment, we also perform an ablation study to evaluate the performance of IMAGENET using different masking techniques. We train IMAGENET using either robust masking through a PGD $\ell\infty$ attack within an $\epsilon := 0.12$ ball, taking 10 steps each of size $\alpha' := 10^{-2}$, or probabilistic masking by sampling features uniformly from the same $\epsilon := 0.12$ ball around $\mathbf{x}$. The average explanation size was $0.07\%$ for probabilistic masking (compared to $0.46\%$ for robust masking), with a faithfulness score of $80.05\%$ (compared to $80.88\%$ for robust masking).

### J.4 IMDB

We now offer more examples of sufficient reasons produced by SST-based models, beginning with cases of IMDB sentiment analysis. As noted in the main paper, subsets derived via probabilistic masking tend to be larger compared to those from baseline masking. This difference arises because the randomness of perturbing various tokens imposes a stricter constraint than simply applying a

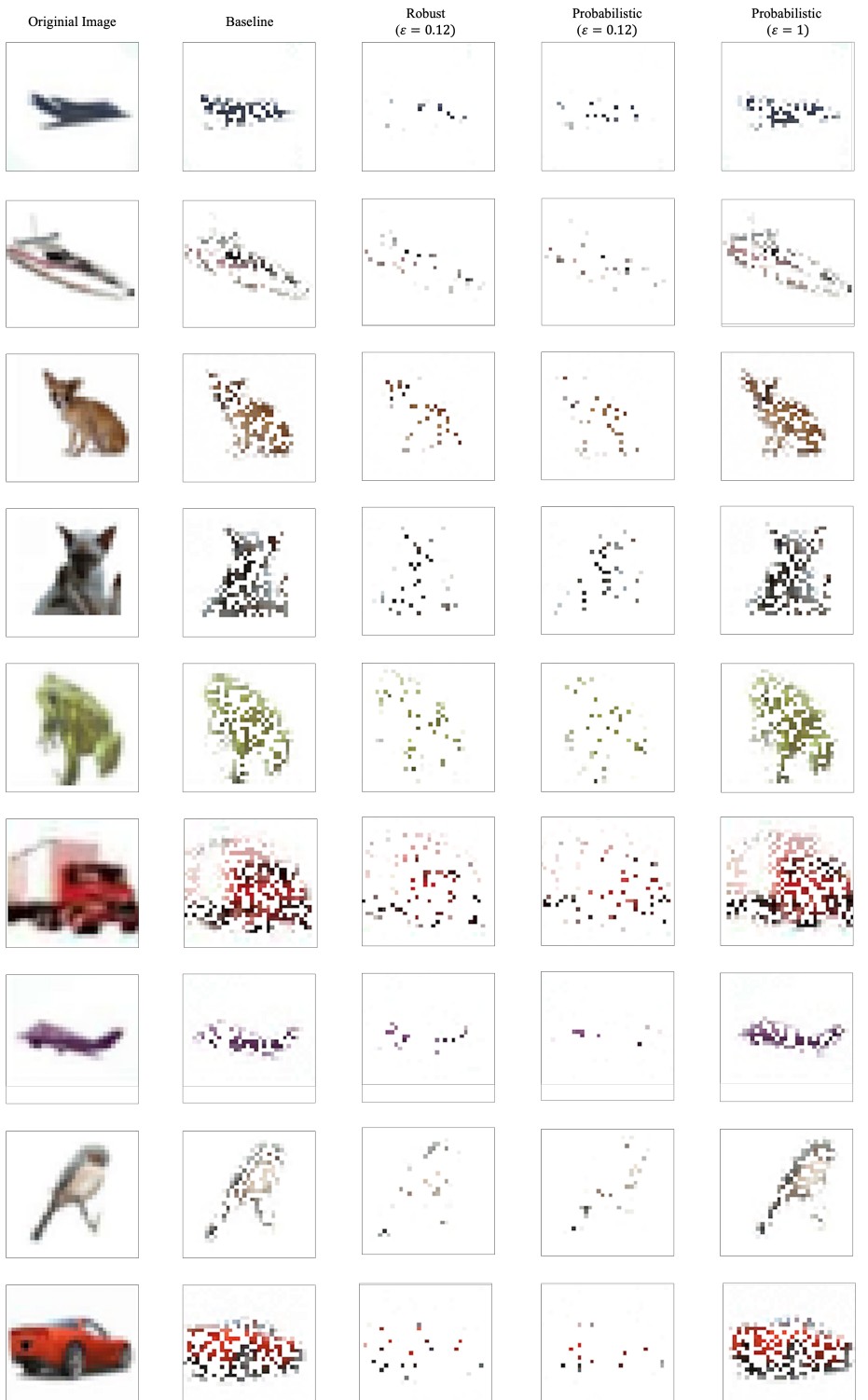

Figure 9: An ablation study comparing between different masking techniques for CIFAR-10

fixed MASK token. Figure 12 provides further comparisons between explanations obtained through probabilistic and baseline masking methods.

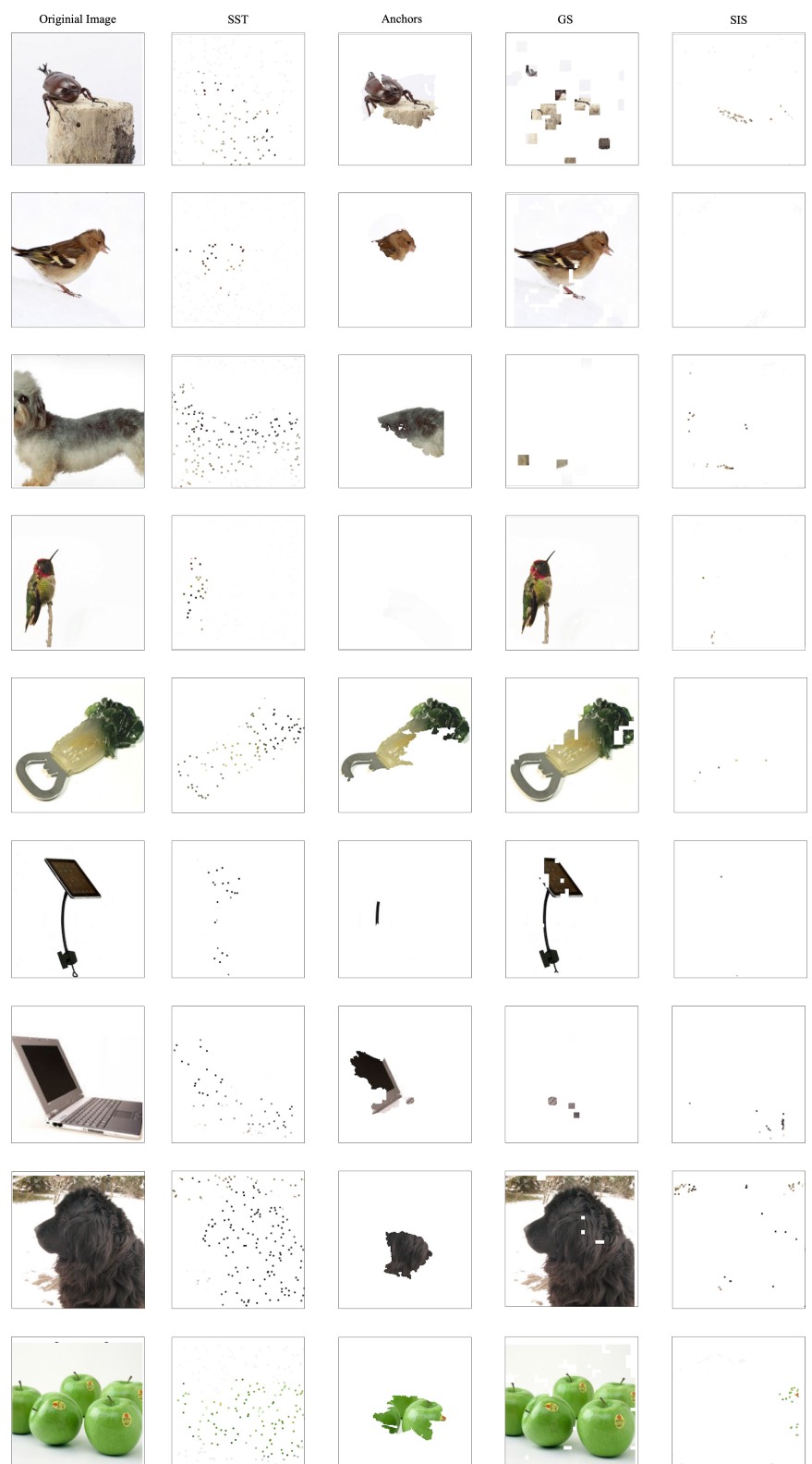

Figure 10: Examples of comparisons between explanations produced by *SST* compared to post-hoc approaches for IMAGENET

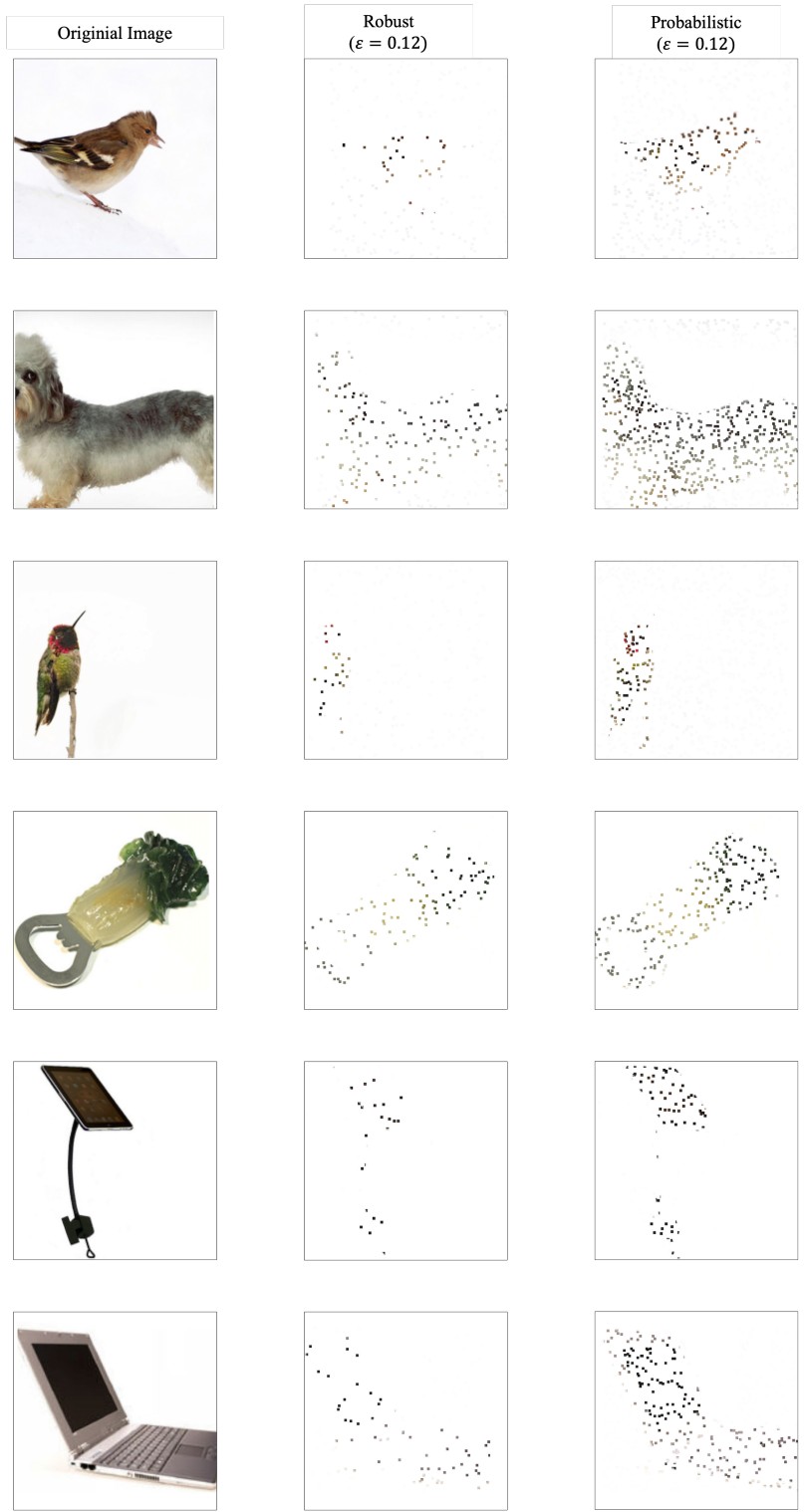

Figure 11: An ablation study comparing different masking techniques for IMAGENET

**Baseline-Sufficiency (negative):**
  "It was terribly *bad* acting … *dumb* *story*.
  *Not* even a kid *would* enjoy … Something to
  *switch* off if possible."

**Probabilistic-Sufficiency (negative):**
  "It was *terribly* *bad* *acting* … dumb *story*.
  *Not* *even* *a* *kid* would enjoy … *Something* to
  *switch* off if possible."

......................................................................................

**Baseline-Sufficiency (negative):**
  "… *Worst* movie *ever*. Mentally- scarred.
  Help me. *It* *was* *that* *bad*. TRUST ME!!!"

**Probabilistic-Sufficiency (negative):**
  "… Worst *movie* *ever*. Mentally- *scarred*.
  *Help* *me*. It *was* that *bad*. *TRUST* *ME*!!!"

......................................................................................

**Baseline-Sufficiency (positive):**
  "It has *everything* a movie should *have*. You
  really *should* *Get* this *one*."

**Probabilistic-Sufficiency (positive):**
  "It has everything a *movie* should have. *You*
  *really* *should* *Get* *this* one."

......................................................................................

**Baseline-Sufficiency (negative):**
  "a *huge* mistake by doing this *film* … The
  best word to describe *it* is *STUPID*!"

**Probabilistic-Sufficiency (negative):**
  "a *huge* *mistake* *by* *doing* *this* film … *The*
  best word to *describe* it is *STUPID*!"

Figure 12: IMDB sentiment analysis sufficient reasons that were inherently generated using SST
(baseline vs. probabilistic)

## J.5  SNLI

Here, we illustrate examples of sufficient reasons generated by SST-based models for the SNLI
classification tasks. In this task, the input contains both a *premise* and a *hypothesis* and the classi-
fication task is to determine the nature of their relationship, categorized as follows: (i) *entailment*,
where the hypothesis is a logical consequence of the premise; (ii) *contradiction*, where the hypothesis
is logically inconsistent with the premise; and (iii) *neutral*, where the hypothesis neither logically
follows from nor contradicts the premise.

**Probabilistic Sufficient reason (contradiction):**
  "*Premise:* A *man* wearing dark *knee* boots, *a* *green* *cape*, and *a* *hat* with *two* big features on
  *the* *front* *is* *sitting*.

  Hypothesis: A man in *a* *red* *cape* and a hat *with* *three* *feathers* *sits* by *himself*."

......................................................................................

**Baseline Sufficient reason (neutral):**
  "*Premise:* old *man* … pink *polo* … with a *small* child *standing* …

  Hypothesis: An *old* *elephant* … through a *telescope* … small child standing next to him.

......................................................................................

**Probabilistic Sufficient reason (contradiction):**
  "*Premise:* … in *a* *black* *leather* shirt *trench*-coat … *faces* away … *visible* just *in* *front* *of* her.

  Hypothesis: … black *leather* *trench* … *walks* *away* … with her *bag* strapped to her *shoulders*.

Figure 13: Sufficient reasons that were inherently generated using SST for the SNLI benchmark

