# OpenReview forum: "Explain Yourself, Briefly! Self-Explaining Neural Networks with Concise Sufficient Reasons"
_ICLR.cc/2025/Conference — ICLR 2025 Poster_

### Official Review · Reviewer_wp7b · 2024-10-30

**Soundness:** 3
**Presentation:** 3
**Contribution:** 3
**Rating:** 6
**Confidence:** 3

**Summary:**

This paper aims at the generation of minimal sufficient reasons for model decision explanation. The authors provide theoretical results on the intractability of getting cardinally minimal explanations under specific settings, showing the difficulty of this problem. The authors also propose a new self-supervised approach called sufficient subset training to reduce cost for generating faithful sufficient reasons and reduce sensitivity to OOD examples.

**Strengths:**

* The paper provides rigourous theoretical results on the computational complexity of generating minimal sufficient reasons in different settings.
* The paper is well-written and easy to follow.
* The proposed SST method provides an elegant and computationally efficient way to generate a sufficient reason. The experiment results show strong scalibility to larger datasets.

**Weaknesses:**

While the paper presents rigorous theoratical results and a novel empirical method, it's a bit unclear to me how these two contributions are connected. Discussion on how the theoratical analysis motivates the SST method could make the paper more integrated.

**Questions:**

1. In Table 2, how is the faithfulness of SST training using different masking strategy in training? It will be interesting to see if model trained with specific masking stretagy also generalizes to other faithfulness metrics.
2. In Figure 4, is Cardinality Mask Size (%) axis refering to the percentage of mask size? According to the figure, cardinality goes to 0.5% instead of 50% as discussed in Line 428-429.

---

> ### Author Response · Authors · 2024-11-17
>
> We appreciate the reviewer’s insightful feedback. Please find our response below.
>
>
> **Improving the connection between the theoretical and empirical aspects of the paper**
>
>
> We agree that the connection between the theoretical and empirical aspects of this paper could be better articulated. Specifically, our findings demonstrate that generating a diverse set of configurations for minimal sufficient reasons is fundamentally intractable. This underscores the potential impracticality of computing such explanations, particularly for large neural networks with expansive input spaces in a post-hoc fashion. Furthermore, our intractability results remain valid even under significantly relaxed conditions, such as approximating the cardinality of the explanations or evaluating sufficiency using only a baseline. These theoretical insights correspond to the observed limitations of many post-hoc methods, which tend to be inefficient and, when applied to larger inputs, often generate subsets that are excessively large or lack faithfulness.
>  This underscores the importance of integrating the learning of sufficient subsets directly during the training phase, thereby eliminating reliance on post-hoc computations and enabling the generation of subsets that are both efficient to generate, concise and faithful. We appreciate your suggestion and will ensure this point is more clearly emphasized in the revised text.
>
>
> **How do different maskings generalize to different forms of sufficiency?**
>
>
> We agree that this is an interesting point to explore. In response to this comment and another from reviewer NPb3, we will include a detailed discussion of this matter in the final draft, along with an additional experiment. We also emphasize that SST can be easily adapted to accommodate several forms of sufficiency by employing diverse types of masking simultaneously during training (a different one at each batch). However, we note that some sufficiency conditions already allow one form to naturally generalize to another, and this is an interesting aspect to discuss. The user who uses SST can determine the choice of sufficiency, and its corresponding masking, and the level of generalizability depends on the relationships between these forms.
>
>
> In Table 2, both robust masking and probabilistic sampling operate within a bounded $\epsilon$ domain, whereas the baseline configuration lies significantly OOD, making its task inherently more challenging. This challenge is evidenced by the larger subset size (23.69%) generated by SST for the baseline compared to the robust and probabilistic settings. As a result, the baseline's larger sufficient reason generalizes well to the other configurations, achieving high faithfulness (98.91% for probabilistic and 98.38% for robust). In contrast, subsets generated by probabilistic and robust masking generalize effectively to each other (98.85% and 99.32%, respectively) but poorly to the baseline (11.82% and 8.16%, respectively), likely due to the baseline's significant OOD nature. This highlights the important role of $\epsilon$ perturbation choice, sampling distribution, and baseline characteristics in influencing the generalization between different maskings.
>
>
> We appreciate you highlighting this interesting point, and we will discuss it more thoroughly in the final version.
>
>
>
>
>
>
> **Additional minor comments**
>
>
> Yes, you are correct regarding the cardinality mask size - this is indeed a typo in the plot. It should be updated from 0.5 to 50. Thank you for catching that!

---

> > ### Comment · Reviewer_wp7b · 2024-11-22
> >
> > Thank you for the response. I'll keep my score.

---

### Official Review · Reviewer_GpEy · 2024-11-03

**Soundness:** 3
**Presentation:** 3
**Contribution:** 3
**Rating:** 6
**Confidence:** 3

**Summary:**

The authors propose a novel training framework called Sufficient Subset Training, aimed at generating minimal sufficient reasons as integral outputs of neural networks.

Unlike post-hoc methods that face computational challenges and out-of-distribution (OOD) concerns, SST directly incorporates the explanation generation process during training.

This is achieved by adding dual propagation and integrating two additional losses: (i) a faithfulness loss for ensuring sufficiency and (ii) cardinality loss for promoting minimal subsets. The method is validated through experiments on various image and language tasks, demonstrating that it provides concise, faithful, and efficient explanations, outperforming several post-hoc methods at finding minimal subset.

**Strengths:**

- The introduction of SST as a self-explaining mechanism is novel and could be impactful for fields focusing on model interpretability.
- The paper thoroughly defines different sufficiency types and explains how SST addresses them through tailored masking strategies.
- The experiments cover a range of datasets and architectures, showing that SST can be applied across different domains.
- The theoretical analysis on the intractability of obtaining minimal sufficient reasons adds depth to the contribution.

**Weaknesses:**

However, I have identified some issues, both major (**M**) and minor (**m**), that require attention.

**M1** Real Practical Insights Are Limited:  while the theoretical framework and empirical demonstrations are solid, the practical insights into what we learn about the internal workings of neural networks are minimal. The method emphasizes explanation generation without significantly advancing our understanding of how or why models arrive at decisions. Addressing this could bridge the gap between theoretical contributions and **practical interpretability**.

**M2** Validity of Faithfulness Metrics: the paper does not thoroughly discuss the potential limitations of the faithfulness metric used in the evaluation. Faithfulness as defined could lead to explanations that superficially appear sufficient but do not align with internal mechanism or key feature used by the model. Say it otherwise, the model may be using completely different strategy to classify x and xS; zS¯.

**M3** Limited Hyperparameter Discussion: the paper mentions tuning the cardinality loss coefficient but provides no analysis on the impact / visual results of tuning the hyperparameters. What are the results of adjusting \( \tau \), step size \( \alpha \) in robust masking. A sensitivity analysis would add valuable context.

Now for the minors (**m**) issues:

**m1** Related Work Section Is Limited:
The related work section lacks depth, with insufficient coverage of recent advancements in XAI and self-explaining methods, especially those published in top venues over the last few years.

**m2** Justification for Masking Strategies:
The choice of certain masking strategies (e.g., baseline and probabilistic) could be better justified. Also, could some baseline lead to better results / scores ? Overall, the rationale for selecting these over alternative approaches should be discussed in more detail.

**m3** Typos and Clarity:
I havent found a lot of typo, just "probablistic" (should be "probabilistic").

**Questions:**

- Could we incorporate some prior like finding mask in a lower dimensional subspace (7x7 grid of the original image) to allow a smoother explanation and more tractable problem for the model ? I think this would be valuable for both interpretability purpose and theoretical verficiation (it become easier to generate gt).
- How does the method's approach compare when applied to more interpretable models (e.g., decision trees)?
- Can the SST approach provide insights into potential biases in training data by analyzing consistent feature patterns?


**Overall, while the theoretical considerations and framework of the paper are strong, the practical interpretability of models remains underexplored. Despite this, the proposed method is an interesting step toward integrating explanation generation within model training, justifying an acceptance with room for further practical development.**

---

> ### Author Response · Authors · 2024-11-17
>
> We appreciate the reviewer’s detailed and insightful feedback. Please find our response below.
>
>
> **Practical implications of the work on interpretability**
>
>
> We thank the reviewer for this comment. While we do believe that our work provides significant contributions in the field, with many practical implications, we agree that our evaluations and analysis focus more on the systematic side of interpretability, emphasizing complexity, faithfulness, conciseness, and efficiency in explanations. However, other practical and human-centered aspects, such as human evaluations, the impact of simplified input settings, relation to bias detection (as noted by the reviewer), remain open for exploration. We will highlight these future directions in the final version.
>
>
> **A further discussion of the validity of the faithfulness metrics should be included**
>
>
> Thank you for this insightful point. We will include a more detailed discussion in the paper and address it in the limitations section as suggested. As with many explanation definitions, the concept of a sufficient reason can be elusive, with the “missingness” of the complement defined in various ways. Our work aims to address this by exploring multiple sufficiency forms. However, we acknowledge that no single definition “perfectly” captures a model’s internal workings, which also applies to faithfulness metrics assessing the sufficiency of subsets. This challenge is common in many XAI methods, such as SHAP [1], other additive attributions [2], and metrics like infidelity [3], where the treatment of “missing” features can vary, leading to differences in explanation definitions and metrics. We believe that the diverse sufficiency criteria explored in our work contribute significantly to a deeper understanding of this type of explanation. In the final version, we will expand the discussion to address the potential limitations of this diversity, which also extend to the faithfulness metrics.
>
>
>
>
>
>
>
>
> **Additional discussion on hyperparameters**
>
>
>
>
> We conducted an ablation study on the cardinality loss coefficient, as it was crucial in demonstrating the inherent trade-off between cardinality and faithfulness in SST, a central element of the framework. To address the reviewers' question about $\tau$ and $\alpha$, adjusting $\tau$ should not fundamentally affect convergence, as the model can adapt to different thresholds. However, extreme $\tau$ values can lead to convergence issues by pushing the model toward subsets that are too large or too small. Hence, we used the default $\tau = 0.5$. As for $\alpha$, as in adversarial training, smaller values discover adversarial examples closer to the input but increase training costs, while larger values find farther examples with lower costs.
>
>
> We agree with the reviewer that more ablations on hyperparameters would be beneficial. In the final version, we will include additional experiments of varying hyperparameters, along with a sensitivity analysis. Thank you for highlighting this point.
>
>
>
> **Can SST be applied to a lower dimensional subspace?**
>
>
> Yes, the reviewer is correct that our approach can be applied to any simplified feature space, such as lower-dimensional subspaces or segmented input representations. This is an exciting direction with great potential, particularly in enhancing the human preference of explanations. In response to this and Reviewer NPb3's comment, we will include an experiment in the final paper demonstrating our method on a reduced, segmented input space and comparing it to the non-segmented setting. We will also emphasize exploring simplified input spaces as a key future research avenue.
>
>
>
>
>
>
> **How does the method's approach compare when applied to more interpretable models (e.g., decision trees)?**
>
>
> That’s an interesting question. Although our focus is on neural networks, obtaining cardinally minimal sufficient reasons is theoretically intractable for other models, including even "interpretable" ones like decision trees [4]. This result is surprising, as it reveals computational hardness in deriving explanations even for simple, interpretable models, though the complexity there is “only” NP-complete, which is less than neural networks ($\Sigma^P_2$-complete, etc.).
>
>
> Since obtaining post-hoc explanations for decision trees is computationally challenging, training self-explaining models is an interesting research direction. However, as decision trees and similar interpretable models are less expressive than neural networks, identifying sufficient reasons for their predictions may be harder for the model to learn. This remains an open question. We agree that extending our work to other model types is a valuable research direction and will highlight this for future work.

---

> > ### Author Response · Authors · 2024-11-17
> >
> > **Can SST provide insights into potential biases in training data by analyzing consistent feature patterns?**
> >
> >
> > Yes, we believe that this is indeed possible. From a theoretical perspective, sufficient reasons are well linked to contrastive/counterfactual explanations [5] and forms of bias detection [6]. For instance, consider an excessively biased case of an adversarial backdoor attack, i.e., a subset of features that systematically alters classification. Any local sufficient reason would identify this backdoor, as excluding it would undermine sufficiency. After training SST, analyzing explanations across inputs could reveal such biases. We agree that this is a very interesting idea for future research. We will address these implications in the final draft.
> >
> >
> > **Additional minor comments**
> >
> >
> > Thank you for bringing these issues to our attention. We will revise the related work section and address the typo you identified. Additionally, we will enhance our discussion of the various choices and outcomes associated with different masking configurations.
> >
> >
> > [1] The many Shapley values for model explanation (Sundararajan et al., ICML 2020)
> >
> >
> > [2] Visualizing the impact of feature attribution baselines (Sturmfels et al., Distill)
> >
> >
> > [3] On the (in) fidelity and sensitivity of explanations (Yeh et al., Neurips 2019)
> >
> >
> > [4] On computing probabilistic explanations for decision trees (Arenas et al., Neurips 2022)
> >
> >
> > [5] From Contrastive to Abductive Explanations and Back Again (Ignatiev et al., KR 2021)
> >
> >
> > [6] On the reasons behind decisions (Darwiche et al., ECAI 2020)

---

> > > ### Comment · Reviewer_GpEy · 2024-11-24
> > >
> > > Thank you for your detailed reply to my comments.
> > >
> > > While I appreciate the authors’ efforts to address the various points raised, I still believe the work overlooks a significant body of literature on attribution methods and their associated metrics. While the theoretical contributions of the paper are undeniable, the practical implications in terms of interpretability remain unclear.
> > >
> > > From a pure XAI standpoint, there is no concrete evidence presented that the explanations provided by the model enhance interpretability in a meaningful way. In particular, the explanations fail to convincingly reveal what the model is doing, which is a critical component of explainability. This leaves me with the impression that the framework may be more suitable for purposes such as certification or auditing rather than advancing explainability.
> > >
> > > That said, I would like to congratulate the authors once again on their strong theoretical contributions and wish them the best of luck with the acceptance process.

---

> > > > ### Author Response · Authors · 2024-11-24
> > > >
> > > > We appreciate the reviewer’s support for the acceptance of our paper and the recognition of its theoretical contributions, (as well as for the great comments!)
> > > >
> > > >
> > > > We note that we adhered to common conventions in the literature [1-4] regarding sufficiency-based explanations, analyzing and evaluating them using the widely used metrics in this context: efficiency, faithfulness, and conciseness. Metrics such as infidelity, commonly used to evaluate additive attribution methods, are not directly applicable here as they are inherently designed for additive forms. Adapting these metrics for sufficiency-based explanations requires significant modifications and, on their own, represent a promising avenue for future research.
> > > >
> > > > We agree that sufficiency-based explanations are particularly well-suited for applications such as model auditing and certification, as explored in prior work [1,5]. By identifying a minimal sufficient subset of input features, one can verify that models are focusing on a set of desired features for their predictions. From the human perspective, methods like Anchors [6] have demonstrated their utility as well. For example, observing multiple explanations for predictions helps humans better predict model decisions compared to relying on additive explanations. We agree that while our work lays the groundwork for addressing the scalability challenges these explanations face, a key avenue for future research lies in enhancing their human-centered aspects. As the reviewer noted, this improvement can enhance the alignment between the theoretical and systematic aspects of these explanations and their human-centered components.
> > > >
> > > > We would like to once again thank the reviewer for their valuable comments!
> > > >
> > > > [1] Verix: Towards verified explainability of deep neural networks (Wu et al., Neurips 2023)
> > > >
> > > > [2] Abduction-based explanations for machine learning models (Ignatiev et al., AAAI 2019)
> > > >
> > > > [3] Distance-Restricted Explanations: Theoretical Underpinnings & Efficient Implementation (Izza et al., KR 2024)
> > > >
> > > > [4] Towards Formal XAI: Formally Approximate Minimal Explanations of Neural Networks (Bassan et al., TACAS 2023)
> > > >
> > > > [5] Auditing Local Explanations is Hard (Bhattacharjee et al., Neurips 2024)
> > > >
> > > > [6] Anchors: High-precision model-agnostic explanations (Ribeiro et al., AAAI 2019)

---

### Official Review · Reviewer_NPb3 · 2024-11-07

**Soundness:** 3
**Presentation:** 3
**Contribution:** 2
**Rating:** 6
**Confidence:** 3

**Summary:**

This paper proposes a new training method called Sufficient Subset training, which proposes learning neural networks with a dual objective that first predicts the label but also predicts a mask of sufficient inputs that is enough to predict the object. They show that training a network this way gives faithful sufficient subset explanations, that are often smaller than those generated with post-hoc methods, and much faster to produce.

**Strengths:**

- Very well written, the description of previous works in terms of 3 definitions is particularly clear and helpful.
- Interesting theoretical results justifying the use of training time intervention
- Clever and relatively simple method
- Some good results, especially on MNIST

**Weaknesses:**

- Decent loss in performance/overall less impressive results on ImageNet and BERT
- Table 2 and 3 missing sufficiency results of your models on the metric it wasn't trained on. Seems unfair to report baselines on all metrics but your methods only under the best metric.
    - I'm worried the self-explanation might overfit to the specific training scenario and not work well in other settings.
- Having very disjoint input sets looks pretty weird for image explanations, what do you think is the use case for this type of explanation? Overall could use some more discussion on why sufficiency explanations are useful/what is the proposed use case. Have you experimented with favoring more continuous regions?

**Questions:**

- What's stopping the model from learning a "cheating" solution such as it almost always only looks at a certain subset of the inputs and this subset is always the explanation? This would probably not be what we want?
- What is the robustness of your different masking strategies on Table 2 and Table 3 when evaluated in a setting not trained on?
- Is the epsilon ball used too small? Theoretically the robust sufficient reasons case should be harder than baseline etc, but seems like you can learn much smaller explanations with this method. As an extreme case, an adversarially robust model could have a sufficient explanation of size 0 and still be faithful in this metric, but this seems pretty detached from the idea of sufficient explanation.

---

> ### Author Response · Authors · 2024-11-17
>
> We appreciate the reviewer’s valuable comments. Please find our response below.
>
>
> **Results on ImageNet and BERT**
>
>
> Although we recognize that the performance drop was indeed slightly more noticeable on ImageNet (as highlighted in the limitations), BERT-based models experienced less than a 1% decrease in accuracy across all benchmarks, while delivering significantly more faithful and notably more efficient explanations.
>
>
> **Why sufficiency explanations are useful, and the use of feature-level explanations**
>
>
> We appreciate the reviewer’s suggestion to enhance the background on sufficient explanations and we will address this in the final version. Sufficient explanations provide a distinct explainability framework compared to the more popular additive attribution methods like SHAP, LIME, or Integrated Gradients, offering insights often overlooked by these approaches, particularly around feature interactions and non-linear behaviors. For example, the top $k$ weighted coefficients in additive attributions do not indicate whether those features alone determine the prediction, a gap that sufficient explanations fill. The authors of Anchors [1] demonstrate that such explanations often offer more intuitive and human-preferred insights than traditional additive ones.
>
>
> We focus on direct sufficient subsets of the input space, aligning with methods tackling the same task [2-5], while offering advantages like: (1) detailed, localized insights, (2) reduced arbitrary segmentation issues, (3) minimized information loss, and (4) improved faithfulness of predictions. While our results capture the minimal input subset for predictions, we agree with the reviewer that extending the framework to higher-dimensional spaces could enhance certain aspects of interpretability, particularly from a human perspective. This opens opportunities for future work. Importantly, our method can be applied to any feature space.
>
>
> In response to this and Reviewer GpEy's comment, we will add an experiment to the final paper applying our method to a reduced, segmented input space and highlight the value of exploring additional simplified input spaces as a valuable direction for future work.
>
>
>
>
>
>
>
>
>
>
> **What’s stopping the model from learning a “cheating” solution that always produces the same subset?**
>
>
> We agree that this is an important point. Like many deep learning tasks, our framework risks the model exploiting "shortcuts" or converging to undesirable local minima. However, our optimization objective explicitly avoids favoring the mentioned configuration, making such convergence very unlikely.
>
>
> First, it is important to emphasize that the explanations generated by our approach are inherently *local* rather than global. For each input, the model identifies a unique sufficient subset specifically tailored to that input. Because different inputs usually require substantially different subsets as minimal sufficient reasons for their predictions, a model that consistently produces the same subset would inherently fail to be faithful, contradicting our objective of optimizing for faithfulness.
>
>
> Furthermore, as shown in the figures in the main text and appendix, subsets generated for different inputs vary significantly within the same benchmark, demonstrating this issue does not arise in practice. We ran an initial experiment on CIFAR10 with robust masking (average explanation size: 12.99%) to support this: 0.07% of pixels appeared in 84% of explanations, 0.14% in 70–80%, 28% in less than 10%, and the remaining 72% varied from 10–80%. While some overlap is observed (which is expected, as the important pixels typically appear near the center of the image), the explanations overall exhibit significant variation.

---

> > ### Author Response · Authors · 2024-11-17
> >
> > **The ability of different masking settings to generalize to different sufficiency configurations**
> >
> >
> > In Tables 2 and 3, we aimed to showcase SST's ability to optimize different masking criteria, demonstrating that it can learn to generate concise and faithful sufficient reasons for any given form of sufficiency. We agree that investigating how different masking configurations generalize to other forms of sufficiency is an interesting point to explore. Following this remark, we will include a detailed discussion and a dedicated experiment in the final version. First, we note that SST can easily be generalized to uphold a set of several forms of sufficiency combined by training with a mix of masking configurations, such as varying masks across different batches. However, some masks already naturally generalize better to others, depending on the form of sufficiency, the baseline, and the input distribution.
> >
> >
> > In Table 3, focused on the language task, we observe the following: For SNLI, probabilistic masking achieves 93.12% baseline faithfulness, while baseline masking achieves only 44.81% probabilistic faithfulness, indicating better generalization under probabilistic constraints. For IMDB, probabilistic masking reaches 77.7% baseline faithfulness compared to 75.7% probabilistic faithfulness for baseline masking, showing both methods generalize moderately well.
> >
> >
> >
> >
> > In Table 2, a more nuanced dynamic emerges. Both robust masking and probabilistic sampling occur within a bounded $\epsilon$ domain, while the baseline configuration is significantly OOD, making the baseline task more challenging. This is reflected in the larger subset size (23.69%) produced by SST for the baseline, compared to the robust and probabilistic settings. Consequently, the baseline sufficient reason, which is larger, generalizes well to the other configurations, maintaining high faithfulness (98.91% probabilistic, 98.38% robust). However, subsets from probabilistic and robust masking generalize well to each other (98.85% and 99.32%, respectively) but poorly to the baseline (11.82% and 8.16%, respectively), likely due to the baseline's significant OOD nature. This underscores the interplay between $\epsilon$ domain choice, sampling distribution, and baseline properties in shaping generalization.
> >
> >
> >
> >
> > We agree that the generalization of different masks across forms of sufficiency is interesting. We will address this more thoroughly, including an additional experiment, in the final version. Thank you for highlighting this!
> >
> >
> >
> >
> >
> >
> >
> >
> >
> >
> > **Is the perturbation in robust masking SST too small?**
> >
> >
> > SST is capable of adapting to various $\epsilon$ perturbations, each representing a different sufficiency configuration. Users can choose different $\epsilon$ values based on the desired "degree" of sufficiency. Intuitively, smaller $\epsilon$ values result in the model identifying a concise subset of features that satisfy this sufficiency level, while larger $\epsilon$ values lead to an increase in the size of the selected important features. The selection of the $\epsilon$ value for our experiments was chosen as it struck a good balance in our experiments: it posed a challenging faithfulness task to prevent the model from converging to a zero explanation size while also avoiding excessive image distortion. We note that although increasing $\epsilon$ raises the "difficulty" of the learned sufficiency form, excessively large $\epsilon$ values can diminish the impact of gradient perturbations, which is a common problem in adversarial training [6].
> >
> >
> > To clarify this point further, our final draft will include an ablation study on $\epsilon$ perturbations and other hyperparameters suggested by reviewer GpEy. We thank the reviewer for this valuable point.
> >
> >
> >
> >
> >
> >
> >
> >
> > [1] Anchors: High-precision model-agnostic explanations (Ribeiro et al., AAAI 2018)
> >
> >
> >
> >
> > [2] What made you do this? understanding black-box decisions with sufficient input subsets (Carter et al., AI’STATS 2019)
> >
> >
> > [3] Abduction-based explanations for machine learning models (Ignatiev et al., AAAI 2019)
> >
> >
> > [4] Verix: Towards verified explainability of deep neural networks (Wu et al., Neurips 2023)
> >
> >
> > [5] Overinterpretation reveals image classification model pathologies (Carter et al., Neurips 2021)
> >
> >
> >
> > [6] Scaling Adversarial Training to Large Perturbation Bounds (Addepalli et al., ECCV 2022)

---

> > > ### Author Response · Authors · 2024-12-02
> > >
> > > Dear Reviewer NPb3,
> > >
> > > Thank you once again for your thorough and insightful feedback, which has been invaluable in highlighting areas of our paper that could benefit from further clarification.
> > >
> > > As the rebuttal period nears its conclusion, we would appreciate knowing if you have any additional questions or concerns that we could address.
> > >
> > > Best regards,
> > >
> > > The Authors

---

> > > ### Comment · Reviewer_NPb3 · 2024-12-02
> > > **Response to rebuttal**
> > >
> > > Thank you for the response!
> > >
> > > This addresses some of my concerns, such as learning of cheating solutions, but still leaves a few, in particular the generalization performance of your method to other settings is not particularly strong and the paper would be stronger with results of a training method that performs well across multiple settings such as a hybrid objective you discussed. In addition a study on the epsilon hyperparameter and similar would make the paper stronger as you are planning, but I find the current experimental results still a little non-convincing and cannot justify raising my score based on proposed experiments so I maintain my initial rating.

---

> > > > ### Author Response · Authors · 2024-12-02
> > > >
> > > > We thank the reviewer for their response and their valuable feedback.
> > > >
> > > > We are happy to know that some of your concerns have been addressed.
> > > >
> > > > Regarding the ablations on various $\epsilon$ perturbations, we acknowledge the value of additional experiments like the one suggested. While our current study includes ablations, such as on the cardinality loss coefficient to emphasize the cardinality-faithfulness tradeoff, we agree that further analysis would be beneficial. That said, similar to adversarial training, robust masking is computationally expensive, making it challenging to perform a comprehensive ablation across all benchmarks within the rebuttal's time constraints.
> > > >
> > > > Nevertheless, we were able to complete some ablation experiments planned for the final version. Here, we present initial results from a study conducted on CIFAR-10:
> > > >
> > > >
> > > > | Perturbation Radius | Explanation Size | Faithfulness |
> > > > |---------------------|------------------|-------------------|
> > > > |               0.01 | 9.02% |  99.33%|
> > > > |               0.05 | 12.93% | 95.63%  |
> > > > |               0.12 | 12.99% |  90.43%  |
> > > > |               0.2 | 22.98% |  85.38%  |
> > > >
> > > > The point raised in our response regarding the increased difficulty of handling larger $\epsilon$ perturbations is supported by these results, as they show that larger perturbations lead to increased explanation sizes and reduced faithfulness. We will incorporate the final and full ablation study in the final version of our work.
> > > >
> > > > We thank the reviewer again for their constructive feedback.

---

> > > > > ### Comment · Reviewer_NPb3 · 2024-12-03
> > > > >
> > > > > Thank you for the response. I will increase my score to 6

---

> > > > > > ### Author Response · Authors · 2024-12-04
> > > > > >
> > > > > > We appreciate the reviewer’s feedback and the increased score. In the final version, we will incorporate the complete ablation along with the relevant results and discussions.
> > > > > >
> > > > > > Thank you again for helping us improve our work.

---

### Author Response · Authors · 2024-12-02

We thank the reviewers once again for their valuable feedback and for recognizing the significance of our work.

We have addressed many reviewer concerns in their respective threads. However, we would like to address two general comments raised by reviewers in the general thread due to their importance.

**Practicality aspects of sufficient explanations**

Minimal sufficient explanations are a widely sought-after approach in explainability, with numerous methods proposed to achieve them (e.g., 1-4). The core idea is to identify a minimal subset of features that determine a prediction, allowing one to focus on the essential "reason" for the outcome while excluding redundant features. Some of the figures and demonstrations in our work clearly highlight that, despite the remarkably small size of our generated subsets, the prediction can often be accurately inferred solely from the subset itself, without needing to consider its complement. This level of faithfulness is not always achieved by post-hoc methods like SIS or approaches that produce significantly larger subsets (e.g., Anchors). Moreover, previous work demonstrated that this form of explanation often provides humans with deeper insights into predictions than additive attributions [4].

We thank reviewers GpEy and NPb3 for their insights on human-centric aspects and enhancements of sufficient explanations, such as simplified inputs and broader use cases. While these are valuable directions, they are not unique to SST but broadly relevant to this explanation type. We believe that our work marks a significant step forward by substantially improving the scalability of generating sufficient explanations while enhancing their faithfulness and conciseness. This advancement lays a strong foundation for future efforts aimed at refining the human-level interpretation of these explanations.


**The generalization of different maskings to different sufficiency conditions and the faithfulness metric**

Like many explanation frameworks, sufficient reasons can be defined in various ways due to the inherent challenge of specifying the "missing" complement of a subset $S$. This issue also appears in Shapley values, which permit multiple definitions [5], and in metrics like fidelity [6]. To address this gap, common in many other explainability approaches, we explored and categorized definitions into three forms: baseline, probabilistic, and robust sufficiency. We demonstrated how various masking configurations can adapt to these forms, enabling users to choose a "form" of sufficiency to guide concise subset extraction. Stricter sufficiency yields larger subsets, while looser forms produce smaller ones.

Naturally, training with a specific masking form tailored to a particular definition of sufficiency may not generalize well to others, similar to how adversarial training guaranteeing robustness against $\ell_{\infty}$ attacks does not ensure robustness against $\ell_0$ attacks. However, as noted in our responses to reviewers NPb3 and wp7b, some sufficiency-based maskings can generalize well, especially when they subsume other forms. Moreover, SST can be adapted to include multiple masking criteria across batches. Lastly, as Reviewer GpEy noted, diverse sufficiency definitions also yield varied faithfulness criteria. To address this, our paper explores multiple forms aligned with these definitions.

In conclusion, while we acknowledge the reviewers' concern that the diverse definitions of sufficiency and faithfulness present challenges - such as generalization and evaluation - we see this as a broader issue in post-hoc explanations rather than a limitation specific to SST. We believe that our exploration of many sufficiency definitions and the demonstration of SST's ability to enhance scalability, faithfulness, and conciseness across distinct sufficiency forms highlights its versatility and applicability.

We will focus on enhancing the discussion of these aspects in the final version and will incorporate additional results, as outlined in the individual threads.

Once again, we thank the reviewers for their insightful feedback.

[1] What made you do this? understanding black-box decisions with sufficient input subsets (Carter et al., AI’STATS 2019)

[2] Verix: Towards verified explainability of deep neural networks (Wu et al., Neurips 2023)

[3] Abduction-based explanations for machine learning models (Ignatiev et al., AAAI 2019)

[4] Anchors: High-precision model-agnostic explanations (Ribeiro et al., AAAI 2018)

[5] The many Shapley values for model explanation (Sundararajan et al., ICML 2020)

[6] On the (in) fidelity and sensitivity of explanations (Yeh et al., Neurips 2019)

---

### Meta-Review · Area_Chair_KQnd · 2024-12-18

**Metareview:**

The paper proposes Sufficient Subset Training, a method for training neural networks to generate sufficient reasons for predictions, which combines dual propagation with faithfulness and cardinality losses, to ensure explanations are concise and faithful. The experiments demonstrate the method's scalability, conciseness, and efficiency compared to post-hoc methods.

The strengths of the paper lie in a novel integration of explanation generation into the training process, empowered with a theoretical analysis of sufficient reasons. While the practical interpretability insights appear to be limited, the merits of the paper outweigh its weaknesses.

I agree with the consensus of the reviewers and recommend accepting the paper.

**Additional Comments On Reviewer Discussion:**

The reviewers engaged with the authors during the rebuttal. The authors were able to address the raised questions convincingly.

---

### Decision · Program_Chairs · 2025-01-22

Accept (Poster)